# Inclusion of flood diversion canal operation in the H08 hydrological model with a case study from the Chao Phraya River Basin: Model development and validation

Saritha Padiyedath Gopalan[1], Adisorn Champathong[2], Thada Sukhapunnaphan[2], Shinichiro Nakamura[3], Naota Hanasaki[1]

[1]Centre for Climate Change Adaptation, National Institute for Environmental Studies (NIES), 16-2 Onogawa, Tsukuba, Ibaraki 305-8506, Japan
[2]Royal Irrigation Department, 811 Samsen Road, Dusit, Bangkok, Thailand
[3]Royal Department of Civil Engineering, Graduate School of Engineering, Nagoya University, Building No. 9, Furo-cho, Chikusa-ku, Nagoya 464-8603, Japan

*Correspondence to*: Saritha Padiyedath Gopalan (pgsaritha@nies.go.jp; charu666@gmail.com)

**Abstract.** Water diversion systems play crucial roles in assuaging flood risk by diverting and redistributing water within and among basins. For flood and drought assessments, including investigations of the effects of diversion systems on river discharge worldwide, the explicit inclusion of these systems into global hydrological models (GHMs) is essential. However, such representation remains in the pioneering stage because of complex canal operations and insufficient data. Therefore, we developed a regionalized canal operation scheme and implemented it in the H08 GHM for flood diversion in the Chao Phraya River Basin (CPRB), Thailand, which is a complex river network with several natural and man-made diversion canals and has been subject to severe flooding in the past, including recent years.

Region-specific validation results revealed that the enhanced H08 model with the regionalized diversion scheme could effectively simulate the observed flood diversion pattern in the CPRB. Diverted water comprises approximately 49% of the annual average river discharge in the CPRB. The simulations further confirmed that the presented canal scheme had the potential to reduce flood risk in the basin by significantly reducing the number of flooding days. A generalized canal scheme with simple input data settings was also constructed for future global applications, providing insights into the maximum level of discharge reduction achievable with diversion of nearly 57% of the annual average river discharge of the CPRB. Overall, the enhanced H08 model with canal schemes can be adapted and applied to different contexts and regions, accounting for the characteristics of each river network by maintaining the basic principles unaltered.

## 1 Introduction

Floods are among the most severe natural hazards, and flooding occurs naturally when river discharge exceeds the channel carrying capacity. Extreme flood events cause serious damage to human life, property, and agricultural systems with disrupted economic activity (Padiyedath Gopalan et al., 2020; Yin et al., 2018). In past decades, annual losses caused by

extreme flood events have been approximately tens of billions of US dollars; moreover, thousands of people have lost their lives in floods each year (Hirabayashi et al., 2013). Currently, almost 13% of the global population (around one billion people) lives in floodplains (Di Baldassarre et al., 2013); this population are at risk of exposure to river flooding caused by extreme weather events (Alfieri et al., 2017). Climate change may increase the intensity and frequency of extreme flood events and the global population exposed to flooding, thereby enhancing the magnitude of losses and fatalities (Dankers et al., 2014; Hirabayashi et al., 2013; Hirabayashi et al., 2021; Jongman et al., 2015; Prein et al., 2017). The current flood issues have been addressed in many areas to a greater extent through the construction of 16.7 million reservoirs with a storage volume of nearly 8000 km$^3$ (Lehner et al., 2011), along with other flood mitigation measures such as retention areas, afforestation, diversion dams, flood channels, etc.

Among different water infrastructures, water diversion systems can be an influential driver for flood management by diverting water from surplus areas to deficit ones in various river basins (ICID 2007). Concurrently, they also play a vital role in the sustainable development of water resources by distributing the diverted water for various purposes like irrigated agriculture, municipal and industrial water supply, etc. (Shumilova et al., 2018). Water diversion schemes account for about 14% of total global water withdrawal (540 km$^3$) in 2005 (ICID, 2005). Recently, McDonald et al. (2014) reported that nearly 184 km$^3$ of water is diverted and moved over a distance of approximately 27000 km annually to meet the water demand in large cities globally. Later, Shumilova et al. (2018) presented the first global inventory of 110 water transfer megaprojects (existing, planned, and proposed) that serve multiple purposes. They revealed that the total volume of water transferred by existing systems was nearly 204 km$^3$/year and would increase to 1910 km$^3$/year after the completion of planned and proposed diversion systems. These findings emphasize the necessity of including water diversion schemes in hydrological models (regional and global), which may otherwise overestimate flood risk and water stress.

Only a few attempts have been made to assess the effects of diversion canals on flood risk reduction although water diversion plays an important role in controlling floods of the water supply areas. For instance, the effectiveness of diversion canals in decreasing floodwater levels was evaluated using the HEC-RAS hydraulic model in Indonesian River Basins (Indrawati et al., 2018; Nugroho et al., 2018). In addition, Wang et al. (2019) examined the impacts of inter-basin water diversion on flood control and drainage processes in a water-receiving lake basin in China using the MIKE model. Subsequently, Mel et al. (2020a) identified the optimal floodgate operation rules for diverting floodwater in the Bacchiglione-Brenta River network in Italy using the 2DEF hydrodynamic model; they also evaluated the applicability of those rules (Mel et al., 2020b). Although the use of hydrodynamic models to simulate floodwater diversion has greatly clarified the effects of such operations on flood risk reduction, these studies have mainly considered the relationships between regional flood characteristics and associated diversion canal operation schemes, which have limited global applicability as well as in other regions. This further signifies the integration of water diversion schemes into global hydrological models (GHMs) to assess their impacts on global and regional flood control that are relatively unexplored.

For impact assessment of human interactions on flood discharge worldwide, their explicit inclusion into GHMs is necessary (Bierkens, 2015; Boulange et al., 2021; Nazemi and Wheater, 2015a; 2015b). GHMs have been progressively updated to include reservoir operations; water withdrawal and supply for irrigation, domestic, and industrial uses; groundwater dynamics; and seawater desalination (Burek et al., 2020; Ehsani et al., 2016; 2017; Hanasaki et al., 2018; Müller Schmied et

al., 2021; Pokhrel et al., 2015; Sutanudjaja et al., 2018). However, the representation of water diversion systems remains in the pioneering stage because of complex canal operations and insufficient data. Among the different GHMs, the H08 GHM is a pioneering model that explicitly considers numerous human interactions with the hydrological cycle (Hanasaki et al., 2008a; 2008b; 2018). The H08 model has been successfully applied at global and regional scales (Masaki et al., 2017; Masood et al., 2015; Mateo et al., 2014). However, floodwater diversion has not been implemented thus far because of the

challenges involved in its implementation. Therefore, the focus of this study was the development and implementation of a diversion canal operation scheme into the H08 GHM that was carefully designed to adapt for future floodwater management in complex river networks worldwide using a case study in the Chao Phraya River basin (CPRB), Thailand.

Thailand is the second-largest economy in Southeast Asia, and 58% of all disasters in that country are due to floods (JICA,

2015). Floods in Thailand have caused 27% of total disaster-related deaths; the greatest proportion can be attributed to the 2011 flood disaster in the CPRB, the largest basin in Thailand. The estimated economic losses from the 2011 flood were approximately USD 46.5 billion (JICA, 2015; Komori et al., 2012; World Bank, 2012). The CPRB has been subject to severe flooding in the past, including in recent years (Komori et al., 2012; Kotsuki et al., 2014; Kure and Tebakari, 2012; Watanabe et al., 2014). Recent studies have evaluated the applicabilities of various adaptation strategies, including

afforestation (Takata and Hanasaki, 2020) and reservoir operation (Padiyedath Gopalan et al., 2021), for reducing flood risk in the CPRB; they concluded that further adaptation measures are needed to address future floods. Although the CPRB has several natural and man-made diversion canals, their adaptation potential remains largely unknown, because these diversion systems have not been integrated into hydrological models. Hence, in this study, flood control was considered as the primary feature of implemented diversion canal scheme because of severe flooding issues in the CPRB, although water diversion

systems could function as both floodways and irrigation channels.

The aim and novelty of this study were the construction of a regional floodwater diversion scheme for the CPRB that is flexible and easily modifiable for future application in any region worldwide. Based on the aforementioned discussions, this study is composed of the following objectives:

(i) Develop a floodwater diversion scheme for the CPRB and enhance the H08 GHM by incorporating this scheme into the model. The water diversion scheme includes diversion canals and retention areas to divert and store water, respectively.

(ii) Validate the effectiveness of the enhanced H08 model for reproducing the water diversion process in the CPRB.

(iii) Analyze the impacts of diversion canals and retention areas on flood management in the CPRB.

## 2 Materials and methods

### 2.1 H08 model

H08 is an integrated global water resources model that consists of six modules: land surface hydrology, river routing, crop growth, reservoir operation, environmental water, and anthropogenic water withdrawal (Hanasaki et al., 2008a; 2008b) to quantify natural as well as anthropogenic water availability and use in the past, present, and future. Each of the modules can run separately with standard spatial and temporal resolutions of $0.5° \times 0.5°$ and one day, respectively, on a global scale. The land surface hydrology module is based on a bucket model (Manabe, 1969) that simulates the energy and water balances at the land surface from forcing data. Runoff, the major output from the land surface hydrology module, is then fed into the river routing module, which accumulates the runoff and outputs as streamflow along the river network from upstream to downstream. This module does not account for the effect of water infrastructures and thereby simulates the natural water cycle in cooperation with the land surface hydrology module. Evapotranspiration, another major product of the land surface hydrology module, is used in the crop growth module to estimate global cropping calendars and crop yields. The reservoir operation module follows operating rules for individual reservoirs based on their primary purpose (i.e., irrigation or non-irrigation water supply). The environmental water module receives streamflow estimates from the river routing module and calculates the river flow that should be maintained in the river channel to support the aquatic ecosystem. Finally, the six modules are coupled to run all processes (including anthropogenic water withdrawal) in an integrated manner. In the coupled model, irrigation water demand and streamflow are utilized to estimate anthropogenic water withdrawal. For a detailed description of the H08 hydrological model, please see https://h08.nies.go.jp/h08/index.html.

### 2.2 H08 model enhancement for flood diversion canals

The latest version of the H08 model enables water abstraction from various water sources (e.g., surface water and groundwater); it explicitly includes groundwater recharge, groundwater abstraction, aqueduct water transfer, local reservoirs, irrigation return flow and delivery loss, and seawater desalination schemes (Hanasaki et al., 2018). These changes transform H08 into one of the most detailed GHMs for attributing different water sources available to human society. The aqueduct water transfer scheme was implemented into the H08 model to provide water supply to the grid cells that are farther from the river channel to meet their water demand (agricultural, industrial, and domestic). If there is a water demand to meet, the scheme assumes that the water could be transferred until the river flow at the aqueduct origin falls below the environmental flow because the information regarding the aqueduct carrying capacity was not available for most cases (Hanasaki et al., 2018). The operation strategy transfers water only when the water demand is positive. However, the aqueduct scheme does nothing for excess water availability (floodwater) and the model still lacks representation of floodwater diversion systems; the implementation of this representation will provide viable adaptation scenarios towards a warming climate.

To overcome this limitation, the H08 model was enhanced by implementing diversion canal schemes. Although the primary objective of diversion canals assessed in this study is flood control, the implemented canal schemes could function as both floodways (offering different routes for excess water flow) and water supply canals (redistributing water from one region to another region that is experiencing water shortage) in future global applications. If flood control is the primary purpose of a canal, the river flow is diverted to the floodway (flood diversion canals) only during the wet season. In contrast, river flow is

diverted to water supply canals during the dry season to supply water for various purposes, including irrigation. The schematic diagram of canal operation for the wet and dry seasons are shown in Fig. 1. For multi-purpose canals, river flow is diverted throughout the year to provide water supply during the dry season and flood control during the wet season. Because this study included flood diversion, water supply, and multi-purpose canals in the implemented diversion scheme, this scheme can be applied to any region based on the prevalence of each canal type and the context in which canal operations

are needed.

The dry season is characterized by low rainfall that causes consequent water shortage for an extended period. To alleviate the water scarcity issues, water supply should be provided from the river channel to the neighbouring areas through diversion canals by preserving the environmental flow in the river channel. Based on this aspect, the operation of diversion canal

systems during the dry season leads to the emergence of three cases concerning the environmental flow ($Q_{env}$) required to be maintained in the river channel, which are expressed as follows:

$$D_{dry} = \begin{cases} Q_{min} & ; if \ Q > Q_{env} \ and \ (Q - Q_{min}) > Q_{env} \\ Q - Q_{env} & ; if \ Q > Q_{env} \ and \ (Q - Q_{min}) < Q_{env} \\ 0 & ; if \ Q < Q_{env} \end{cases} \tag{1}$$

where $D_{dry}$ is the daily water diversion during the dry season; $Q_{min}$ is minimum flow diversion; $Q_{env}$ is the environmental flow requirement; and $Q$ is the daily river discharge at the origin of diversion. The first two cases in Eq. 1 represent two low

flow scenarios (Supplementary Fig. S1a and b) and can be explained as follows: (i) the operation strategy tries to meet minimum flow diversion ($Q_{min}$) on the premise of guaranteeing the environmental flow ($Q_{env}$) in the river channel even after water diversion due to enough water availability, and (ii) the diversion criterion attempts to divert water ($Q - Q_{env}$) that is smaller in quantity when compared to the $Q_{min}$ to ensure the required the environmental flow requirement in the river channel due to the relatively low water availability. Using these diversion criteria, environmental flow is maintained in both

cases. No flow is diverted to the canal if river discharge is lower than environmental flow, depicted as the third case in Eq. 1. After the diversion into canals, a portion of the diverted water (either a fixed percentage or based on the water demand including irrigation) is supplied to nearby areas ($Q_{sup}$) along its flow route to meet the water supply needs of that areas until flow either diminishes to zero or reaches its destination. The surplus water supply that was not completely utilized for irrigation ($Q_{sup} - Q_{irr}$) is returned to the main channel. The remaining canal flow, if any, is returned to the canal's

destination, as shown in Supplementary Fig. S1a and b.

The wet season is the period during which most of the annual rainfall is received. This high rainfall will eventually cause flooding in the neighbouring areas whenever river discharge exceeds the river channel carrying capacity. The diversion canals can divert this floodwater from the river channel and restore the river water level below the carrying capacity. From this perspective, five relevant cases can be identified for the operation of diversion canals during the wet season, as shown below:

$$D_{wet} = \begin{cases} Q_{cancap} & ; if\ Q > Q_{rivcap}\ and\ (Q - Q_{cancap}) > Q_{env} \\ Q - Q_{env} & ; if\ Q > Q_{rivcap}\ and\ (Q - Q_{cancap}) < Q_{env} \\ Q_{min} & ; if\ Q < Q_{rivcap}\ and\ (Q - Q_{min}) > Q_{env} \\ Q - Q_{env} & ; if\ Q < Q_{rivcap}\ and\ (Q - Q_{min}) < Q_{env} \\ 0 & ; if\ Q < Q_{env} \end{cases} \tag{2}$$

where $D_{wet}$ is the daily water diversion during the wet season; $Q_{cancap}$ is the maximum canal carrying capacity; and $Q_{rivcap}$ is the river channel carrying capacity. The first four cases in Eq. 2 are based on whether river discharge ($Q$) is greater or smaller than the river channel carrying capacity ($Q_{rivcap}$). The first two out of the four cases correspond to the flood flow scenario, where $Q > Q_{rivcap}$ (Supplementary Fig. S1c and d). In the flood flow scenario, the operation strategy tries to divert the maximum possible amount of floodwater that is equivalent to the $Q_{cancap}$ (Case I of Eq. 2) to keep the river flow below the $Q_{rivcap}$. However, at some point, the river flow in the main channel falls below the $Q_{env}$ after the diversion of $Q_{cancap}$. In such instances, the remaining river discharge after meeting the environmental flow requirement ($Q - Q_{env}$) is diverted into the canal instead of diverting the $Q_{cancap}$ (Case II of Eq. 2). The last two out of the four cases represent non-flood flow scenario, where $Q < Q_{rivcap}$ (Supplementary Fig. S1e and f). Although the river discharge lies below the $Q_{rivcap}$ in the non-flood scenario, a minimum flow ($Q_{min}$) is diverted from the main channel to reduce flooding at downstream locations (Case III of Eq. 2). If the diversion of $Q_{min}$ to the canal reduces the river discharge below the $Q_{env}$, then a decreased quantity of water ($Q - Q_{env}$) is diverted from the main channel rather than the $Q_{min}$ to maintain the environmental flow in the river (Case IV of Eq. 2). Canal diversion remains zero during periods of environmental flow (Case V of Eq. 2). The diverted canal water will drain into retention areas ($Q_{ret}$), if present (Supplementary Fig. S1c–f). Excess water from retention areas will move toward the destination when the storage capacity of the retention areas becomes full. These retention areas or ponds store water temporarily to reduce the flood peak; thus, they extend flow duration (Tourment et al., 2016). At the end of each year, retention storage is assumed to evaporate ($E$) fully and retention storage at the beginning of the following year is assumed to be zero.

Flow that is diverted from the main channel into canals eventually splits into four major components, as shown in Fig. 2. The first component is "returned discharge" (A), which is the canal flow returning to destinations such as the downstream section of the same river or nearby rivers or tributaries within the basin. This returned discharge increases flow at the destination and reduces flow at the origin. The second component is "supply to near grids" (B), which is the portion of the diverted water supplied to each of the grid cells through which the canal passes as well as to the immediate lateral neighbouring grid cells of

the canal. The immediate lateral neighbouring grid cells are decided based on the presence of croplands in these grid cells. This "supply to near grids" component is enabled only during the dry season to meet the water demand. For simplicity, 10% of diverted water is supplied to each of the nearby grid cells in this modelling approach that was further utilized for irrigation. If the demand is less than the water supplied to the local grid, then the surplus water after meeting the demand is added to the discharge of the corresponding grid cell. This river discharge finally returns to the river channel as shown in Fig. S1a and b and thereby closes the water balance. The remaining diverted water after supply will move to the subsequent downstream grid cells. This process is repeated until the diverted flow is fully depleted or reaches its destination. At some point, the diverted water flows to nearby river basins or drains into the sea. This component is designated as "flow out of the basin" (C); it alters the water balance of the river system in question and should be included in water budget analysis to avoid unexplained water imbalance. A portion of the diverted canal flow drains into the retention areas, and the floodwater stored in these areas is called the "retention storage" (D). Once the retention areas get full, the remaining canal flow will move to the next grid. This process is repeated in each grid cell along the canal flow route until flow either diminishes to zero or reaches its destination (either within the basin or out of the basin). The storage of diverted floodwater in retention areas is allowed only during the wet season to supplement flood control capacity alongside diversion systems. In addition to the diverted water storage during the wet season, the retention areas are modeled in such a way that they receive runoff generated from precipitation in each grid based on their areal fraction during both dry and wet seasons. This runoff constitutes a part of retention pond storage and thereby affects the land surface processes. Only the remaining storage capacity is available for the storage of diverted floodwater during the wet season. All these components can be customized (enabled/disabled) for each basin under consideration.

## 2.3 Input data for canal operations

To conduct canal operations, the following data are needed:

(i) Geographic location of the canal: The geographic locations of the canal's origin and destination are needed to identify the donor and recipient systems; this is the minimum information required to proceed further. In addition, the location of the entire canal route enables the identification of areas receiving water from the canal system.

(ii) Location and areal extent of retention areas: The geographic locations of possible retention areas (e.g., low-lying areas along river and canals, ponds, wetlands, etc.) and their areal extents available for storage of floodwater.

(iii) River channel carrying capacity: The carrying capacity of the river channel at the origin of the canal diversion should be obtained. When river discharge at the diversion point is greater than the carrying capacity, water is diverted into the canal.

(iv) Canal carrying capacity: The carrying capacity of the canal should be noted; this represents the maximum flow that can be diverted under flood conditions.

(v) Minimum flow diversion: A minimal amount of flow is diverted from the river channel to reduce downstream flooding under non-flood conditions in the wet season and supply water during the low flow conditions in the dry season.

**Commented [PGS1]:** Reviewer 1, Comment 3
According to the comments, added explanations on how the immediate neighbouring grid cells are decided.

(vi) Days of diversion per year: Regional wet and dry seasons must be identified to enable the diversion of water under flood, non-flood, and low flow conditions.

**2.4 Canal operation schemes**

A canal diversion scheme that utilizes the observed values of all input data (described in section 2.3) specific to a particular region is described hereafter as the regionalized canal scheme. However, when the model is applied to data-scarce regions

and periods, the implementation of the diversion system using the input data described in section 2.3 becomes strenuous. In addition, when this diversion scheme is implemented in GHMs for global analysis, extensive data are needed for each river basin, further hampering the application of GHMs. To overcome these difficulties and support the future global application of the canal diversion scheme, the H08 model was tested with simple input data settings that can be derived from river discharge alone (hereafter designated as the generalized canal scheme).


The generalized canal scheme is regarded as the preliminary survey for future global applications. In the generalized scheme, the geographic location of the canal, as well as the locations and areal extents of retention areas, should be estimated for the study region in a manner similar to the regionalized scheme. Information regarding canal systems can be collected from peer-reviewed articles, official websites, reports from governmental and non-governmental organizations, and newspapers

(Shumilova et al., 2018). Based on this information, the geographic locations of the canal origin, flow path, and destination can be extracted using the Google Earth application. Similarly, low-lying areas along rivers (floodplains) and canals, ponds, and lakes can be recognized as retention areas, if present.

The input data for river carrying capacity, canal carrying capacity, and minimum flow of diversion is difficult to collect for

all canal systems to support the global application of the model. The river channel carrying capacity ($Q_{rivcap}$) is the maximum flow that a channel cross-section can accommodate. When river discharge exceeds $Q_{rivcap}$, flooding occurs. This relationship indicates that the $Q_{rivcap}$ at any station is closely associated with the high flow of that cross-section. Therefore, the Q$_5$ value (the 95-percentile flow, which was equalled or exceeded for 5% of the flow record – a high flow representation) derived from the simulated pristine flow was used in the generalized scheme to represent $Q_{rivcap}$. Flow values above Q$_5$ will

create flood flow conditions at specified locations, thus requiring the diversion of water from the main channel to reduce flood risk. Additionally, the Q$_{50}$ value (the 50-percentile flow, which was equalled or exceeded for 50% of the flow record – a medium flow representation) is used to illustrate the canal carrying capacity that supports the diversion of the approximate long-term mean daily flow in the generalized scheme. This flow level ensures that a substantial amount of water is diverted to the canal system during floods. Lastly, the minimum flow diversion is represented by the Q$_{90}$ value (the 10-percentile

flow, which was equalled or exceeded for 90% of the flow record – a low flow representation). The number of days of

diversion is based on the lengths of the wet and dry seasons; therefore, region-specific data are needed under both diversion schemes.

## 3 Study area and model simulations

### 3.1 Study area

The diversion canal system was developed for the CPRB, which is the largest river basin in Thailand, with an area of approximately 158,000 km$^2$ that covers more than one-third of the total area of Thailand (Fig. 3). The CPRB is home to about 30 million people (40% of the country's population) and includes the capital city, Bangkok, which is located at the Chao Phraya River delta. Bangkok contains 50% of the basin's population; generates almost 80% of the basin's gross domestic product; and is the political, commercial, industrial, and cultural hub of Thailand (Bond et al., 2018).


The CPRB is divided into an upper and a lower basin at Nakhon Sawan, (C.2 station) as shown in Fig. 3 (Komori et al., 2012). The upper basin includes four major tributaries (i.e., Ping, Wang, Yom, and Nan) that have headwaters originating from the northern part of the country. The Wang joins with the Ping River and the Yom joins with the Nan River; the subsequent confluence of the Ping and Nan Rivers at Nakhon Sawan is the beginning of the Chao Phraya River. From

Nakhon Sawan, the river flows to the lower basin through Bangkok and finally drains into the Gulf of Thailand (Tebakari et al., 2012). Other tributaries (i.e., Pasak and Sakae Krang Rivers) join the Chao Phraya River in the lower basin.

Thailand experiences two seasons: a wet season (May–October) and a dry season (November–April). The rainfall distribution over the basin significantly varies, ranging from 1,000 to 2,000 mm. Nearly 90% of total annual rainfall is

received during the wet season, leading to increased flood risk (Bond et al., 2018). The lower basin and downstream parts of the Yom basin have gentle slopes (Fig. 3) and reduced channel carrying capacity; this makes the region highly prone to flooding (Komori et al., 2012). The severe flooding that occurred during the 2011 monsoon season inundated the lower basin and downstream parts of the Yom and Nan Rivers (Mateo et al., 2014), causing over 800 deaths and substantial economic losses of USD 46.5 billion (World Bank, 2012). To minimize flooding caused by reduced channel carrying capacity and

gentle slopes at various locations, the Thai government has implemented measures to protect the CPRB by constructing dams and numerous diversion canals (both natural and man-made).

### 3.2 Water infrastructures in the CPRB

#### 3.2.1 Canal system

The CPRB is a complex river network with several natural and man-made diversion canals, as shown in Fig. 4. Fig. 4 also

shows the observed river and canal carrying capacities at various locations in the CPRB (right-hand side) that are solely

determined by their cross-sections. The carrying capacity values were collected by literature review (JICA, 2013; Tamada et al., 2013) as well as from the Royal Irrigation Department, Thailand. Whenever the river discharge exceeds river carrying capacity, flooding will occur. Therefore, a certain amount of floodwater should be diverted from the river to the canal systems, subject to a maximum value of canal carrying capacity. These canals are either for flood diversion or multi-purpose;

they were implemented to mitigate extreme flood events, protect high-density business and residential areas, and provide irrigation water supply. Water diversion into these canal systems is controlled by barrages installed on the Chao Phraya River. Based on the literature review (JICA, 2013; Tamada et al., 2013) and maps collected from the Royal Irrigation Department (RID) of Thailand, eleven canal systems were identified in the CPRB with direct intakes from the main river channel (Fig. 4). Details of these canal systems are provided in Table 1. Of these eleven canal systems, two are in the Yom

basin; flooding is common in that basin because (i) large-scale flow regulating structures are absent, (ii) the downstream area exhibits a gentle slope, and (iii) various locations have low channel carrying capacities. To alleviate these issues, two diversion canals have been implemented to divert water from the Yom River to the Nan River, downstream reaches of the Yom River, and nearby low-lying paddy fields during floods (Fig. 4a).

The remaining nine canal systems are in the lower CPRB, where flood hazards are frequent because the channel's carrying capacity progressively decreases from Nakhon Sawan (C.2 station) to Ayutthaya (C.35 station) as shown in Fig. 4b. To reduce downstream flooding near Ayutthaya, the Chao Phraya diversion dam on the Chao Phraya River allocates water to multiple canals (i.e., Chainat-Pasak, Makham Thao-Uthong, Tha Chin, Noi, and Chainat-Ayutthaya), immediately upstream of the diversion dam (canals three to seven, respectively, in Fig. 4b). Among these five canals, the Tha Chin River is a major

distributary of the Chao Phraya River that flows through Bangkok and drains into the Gulf of Thailand. This river is considered as a canal system because of its ability to carry floodwater away from the Chao Phraya River. Similarly, the Noi River, which diverts water from the main channel, is considered as a diversion canal. Together, these five canals limit flow to approximately 2840 m$^3$/s, which is the observed river carrying capacity at the Chao Phraya diversion dam (Fig. 4b). Downstream of the Chao Phraya Dam, the Lopburi River splits off from the Chao Phraya River after receiving surplus water;

it then joins the Pasak River. The Lopburi River has one short tributary, the Bang Kaeo River, which originates from the Chao Phraya River. These two canal systems together control floods between Sing-Buri and Ang-Thong. Finally, the Phong Pheng and Bang Ban canals divert water from the main channel to reduce the flow at Ayutthaya; they then join the Noi River, which subsequently recombines with the Chao Phraya River at Bang-Sai.

Four canal systems in the basin are solely responsible for flood control: the Yom-Nan diversions in the upper CPRB and the Lopburi and Bang Kaeo canals in the lower CPRB. The remaining seven canals in the lower CPRB serve both flood control and irrigation water supply purposes. Among all canals in the basin, two remove water from the basin: the Makham Thao-Uthong (providing irrigation water to areas that lie outside the basin) and Tha-Chin (emptying into the Gulf of Thailand, similar to the Chao Phraya River) canals. The diversion of river discharge from the main channel to canals is controlled by

**Commented [PGS2]:** Reviewer 1, Comment 2
Authors would like to clarify that the observed carrying capacities were not computed by the authors.

the operation of diversion dams and other water regulation structures (names of structures are provided in Table 1). However, the gravity transfer of water occurs at the origins of the Phong Pheng and Bang Ban canals.

Timely diversion of flood flows from the main channel is essential for controlling flooding downstream of the lower CPRB, which is a flood-prone area affected by strong tides. Therefore, two questions arise at the time of diversion: when to divert

and how much water to divert. Whenever river discharge exceeds the channel carrying capacity at the origin of a diversion canal, an amount no greater than the canal carrying capacity should be diverted to control flooding. Furthermore, to enable irrigation water supply because canals in the CPRB are solely used to provide water for irrigation and not for other sectors, a minimum flow amount is diverted during the dry season. Table S1 shows the observed river channel carrying capacity, canal carrying capacity, and minimum flow of diversion at the origins of the eleven canal systems (Tamada et al., 2013).

**3.2.2 Reservoirs**

Several dams have been constructed in the CPRB since 1950 to store water from the wet to the dry season, providing flood control and water supply (Bond et al., 2018; Tebakari et al., 2012). Therefore, the Chao Phraya River is highly regulated by existing reservoirs. The GRanD global dam database (Lehner et al., 2011) lists 39 dams in Thailand, in which Bhumibol and Sirikit are the two major reservoirs in the Chao Phraya River. Together, the Bhumibol and Sirikit control approximately 22%

of runoff from the basin, with a combined storage capacity of 23 billion $m^3$ (Bond et al., 2018; Komori et al., 2012). There are six other large-scale reservoirs in the Chao Phraya River system, each with a storage capacity greater than 0.1 billion $m^3$. Detailed information regarding the reservoirs along the CPRB is provided in supplementary material S2.2; reservoir locations in the basin are shown in Figure 4.

**3.3 Input data sources and assumptions for canal operation in the CPRB**

Prior to beginning model setup and simulation, the input data required for simulations in the CPRB were finalized for both regionalized and generalized canal schemes, as follows:

(i) Geographic location of the canal: The origin, destination, and flow path of each canal system were identified using the Google Earth application and digitized into the H08 model. Both regionalized and generalized schemes employed the same geographic location data for simulations.

(ii) Location and areal extent of retention areas: In the H08 model, land grid cells are divided into four sub-cell types: double-crop irrigated, single-crop irrigated, rainfed cropland, and non-cropland (Hanasaki et al., 2008a). Irrigation water is applied to the irrigation sub-cells to maintain appropriate soil moisture during the cropping period, while no irrigation is supplied to rainfed and non-cropland sub-cells (Hanasaki et al., 2018). In most Southeast Asian countries, rainfed cropland comprises low-lying paddy fields that are natural floodplains and cannot be cultivated during the rainy season

because of flooding (Jamrussri et al., 2018). Thus, for the sake of brevity, the areal fraction of rainfed croplands (Siebert et al., 2010) presumedly located near the canal route was regarded as the areal fraction of retention areas for both canal

schemes because our region of interest is in a Southeast Asian country. The retention areas modeled under the regionalized scheme were then refined by matching the areal extent of rainfed croplands near the canal networks with the data provided by the RID (geographic location and areal extent of several lowland areas, such as paddy fields, that are currently used as flood retention areas in the basin).

In both schemes, the area of retention ponds in each grid cell was obtained by multiplying the area of that grid cell with the areal fraction of rainfed cropland. Then, this area was multiplied with the depth of water storage (assumed to be one m in this study based on recommendations from the RID) to calculate the storage capacity of the retention ponds. The diverted canal flow was assumed to be temporarily stored in these rainfed croplands until their storage capacity was reached. Further details regarding the inclusion of canal systems and retention areas in the H08 model are provided in supplementary material S3.

(iii) River channel carrying capacity: A slightly adjusted version of the observed channel carrying capacity was used for the regionalized canal scheme. Because the simulated streamflow is slightly underestimated, the river carrying capacity was adjusted to avoid the overestimation of canal effects. This adjustment was conducted by matching the simulated and observed fractions of river flow that are diverted into the canals during the wet and dry seasons. In contrast, $Q_5$ values derived from the simulated pristine flow levels at various stations were used for the channel carrying capacity in the generalized canal scheme. Further details are provided in supplementary material S3.

(iv) Canal carrying capacity: Similar to the river carrying capacity, a slightly adjusted version of the observed canal carrying capacity was used in the regionalized scheme, while $Q_{50}$ values represented carrying capacity in the generalized scheme.

(v) Minimum flow diversion: For the regionalized scheme, the minimum flow to be diverted was obtained from historic canal operation data provided by the RID; these flow data were then adjusted. Under the generalized scheme, these data comprised $Q_{90}$ values, as explained in section 2.4.

A detailed comparison of the two schemes in terms of input data is provided in Table 2 and the corresponding values are given in Table S1.

### 3.4 Model setup and simulations

We previously adapted the H08 model to the calculation domain of 13–20°N and 97–102°E at a spatial resolution of five arcmins for application in the CPRB (Padiyedath Gopalan et al., 2021). In that model setup, only three modules (land surface hydrology, river routing, and reservoir operation) of the H08 model were enabled. Four key parameters of the land surface hydrology module were derived for the CPRB region by calibrating the model at Nakhon Sawan, while generic operation rules of reservoirs in the global setup were improved by including release rates for the wet and dry seasons, as well as an upper rule curve based on observed historical reservoir operations in the CPRB. The simulations were conducted using the IMPAC-T dataset (Kotsuki et al., 2014), which comprises seven meteorological variables including temperature, specific

humidity, short-wave radiation, long-wave radiation, atmospheric pressure, wind speed, and precipitation from 1980 to 2004.
In that study, river discharge and dam operation in the basin were successfully validated.

For simulations in the present study, the same model setup for the CPRB was maintained, except for model calibration. The H08 model was recalibrated for the CPRB because the present study uses an upgraded version of the H08 model that includes four additional parameters (a total of eight parameters) in the land surface hydrology module, representing
groundwater recharge and abstraction (Hanasaki et al., 2018). The eight key parameters of the land surface hydrology module were derived through calibration of the model at Nakhon Sawan. The calibrated parameters and their values are shown in Table S4. Furthermore, five types of discharge simulation were conducted (Table 3) from 1980 to 2004 (total duration of 25 years) as follows: (i) naturalized simulation (NAT), where all types of water infrastructure were disabled; (ii) dam simulation (DAM), where the reservoir operation module was enabled; (iii) irrigated simulation (IRG), where both the
reservoir operation module and irrigation water abstraction module were enabled; (iv) regionalized simulation (REG), which is irrigated simulation with the regionalized canal scheme enabled; and (v) generalized simulation (GEN), which is irrigated simulation with the generalized canal scheme enabled. Equilibrium for all model state variables was achieved using spin-up calculations.

Initially, model calibration and validation were conducted using the pristine flow, and the efficacy was evaluated based on Nash-Sutcliffe efficiency (NSE; Nash and Sutcliffe, 1970). For model validation, daily river discharge data at various locations in the CPRB (green colour gauging stations in Fig. 4) were collected from the RID. In addition, the capability of the H08 model to reproduce observed discharge at Nakhon Sawan using the calibrated parameters was examined. Regionalized canal operations were calibrated for each diversion canal system to mimic the available observed data provided
by the RID. Calibration was conducted by controlling the fraction of river flow to be diverted into the canals during the wet and dry seasons, with reference to the observed fraction of water diverted. Although the wet season ends in October (section 3.1), May–December was regarded as the wet season in this study based on observed reservoir release (Padiyedath Gopalan et al., 2021) and canal diversion operations. Similarly, the dry season was redefined as January–April for simplicity of simulation.

**4 Results**

The H08 model was calibrated for pristine flow at Nakhon Sawan and further validated at various stations in the CPRB. In addition to the pristine flow, the ability of the calibrated parameters in reproducing the observed discharge at Nakhon Sawan was assessed; the results are shown in supplementary material S4.

**4.1 Validation of seasonal water diversion**

The observed and simulated fractions of river flow diverted into the canals with respect to a reference gauging station are listed in Table 4 for the wet and dry seasons. For each canal system, the nearest upstream gauging station was used as the reference gauging station (Column two of Table 4; Fig. 4) to determine the fraction of water diverted. The REG simulation closely agreed with the observed diverted fraction for all canal systems considered during both dry and wet seasons. In the GEN simulation, the diverted fraction was reasonably reproduced at many stations; it substantially differed from the

observations at some stations during both seasons. On average, the REG and GEN simulations diverted 8.78% and 22.55% of river discharge, respectively, compared with the observed dry-season discharge diversion of 9.25%. The average values of observed, REG, and GEN diverted fractions during the wet season were 11.24%, 11.90%, and 11.71%, respectively.

Simulated monthly mean canal flows (REG and GEN) at eleven canal origins are shown in Fig. 5, along with observed

diverted canal flow. Monthly mean values were used to validate canal operations because observed canal operation data are available for a different period from the simulation time frame (1980−2004). The REG canal flow simulation showed good agreement with the observed flow at all stations except the Lopburi and Bang Kaeo canals. The observed data at these two stations included missing values, which reduced the predictive accuracy of flow diversion. From a seasonal perspective, the model exhibited good reproduction of canal flows during both dry and wet seasons. However, the GEN canal flow

simulation exhibited disparities compared with the observed flow at Makham Thao-Uthong, Tha Chin, Chainat-Ayutthaya, Lopburi, Bang Kaeo, and Phong Pheng stations. These disparities can be attributed to uncertainties in the $Q_5$, $Q_{50}$, and $Q_{90}$ values assigned to the channel carrying capacity, canal carrying capacity, and minimum flow diversion, respectively; they can also be attributed to missing values at the Lopburi and Bang Kaeo stations.

The H08 model accurately reproduced canal operations in the CPRB under the REG canal scheme. Although the primary purpose of the implemented canal scheme was flood control during the wet season (May–December), the model was also able to replicate the diversion pattern during the dry season (January–April). This result reveals that the REG canal scheme can also be successfully used to provide water supply during the dry season in addition to the flood control during the wet season.

**4.2 Breakdown of the diverted canal flow**

The four components of diverted canal flow are returned discharge, supply to near grids, flow out of the basin, and retention storage, as described in section 2.2. Fig. 6(a) and (c) show the amount of flow transferred from the canal systems to each of these components under the REG and GEN schemes, respectively, from 1980 to 2004. The maximum and minimum values of diverted annual canal flow were approximately 22 and 7 km$^3$ in the years 1980 and 1991, respectively, under the REG

scheme. These values were similar to the maximum and minimum values obtained from the GEN scheme (21 km$^3$ in 1980

and 9 km$^3$ in 1991). This consistency reveals that the GEN scheme could predict the annual flow diversion pattern in the basin. Fig. 6 shows that most diverted water is gradually returned to the destination points under both canal schemes. The supply to near grids and flow out of the basin were nearly equivalent in the REG scheme, whereas flow out of the basin was dominant in the GEN scheme. This dominance in the GEN scheme can be attributed to the high carrying capacity of canals that flow out of the basin. The last component, retention storage, was small and comparable under both schemes.

Although the total flow diverted to the canal systems was similar under both schemes, the percentage of canal flow transferred to each component differed, as shown in Fig. 6(b) and (d). The annual average flow diversion from the river channel to the canal system under the REG scheme was nearly 13 km$^3$/year, which constitutes nearly 49% of the annual average river discharge in the CPRB (26.6 km$^3$/year). The annual average flow diversion under the GEN scheme was approximately 15 km$^3$/year (57% of the annual average discharge in the CPRB). The share of canal flow attributed to each component was in the following order under the REG scheme: returned discharge, supply to near grids, flow out of the basin, and retention storage. Under the GEN scheme, flow out of the basin exceeded supply to near grids, as described above. Retention storage was 3.2% of total diverted canal flow (415 MCM) under the REG scheme and 649 MCM under the GEN scheme (4.3% of the total diverted canal flow), out of the total storage volumes of 615 MCM (REG scheme) and 935 MCM (GEN scheme; Table S3). Therefore, nearly 67% and 69% of the retention storage was utilized to store floodwater in the REG and GEN schemes, respectively.

In practice, water stored in low-lying retention areas of the CPRB (paddy fields) drains into canals and rivers after the flood, thereby creating space for water from upcoming floods in the same year. This process was not considered in the simulations, and the retention areas were assumed to accommodate floodwater only once per year until they became full. Because of this simplification, neither diversion canal scheme was able to reach the observed potential storage capacity (1701 MCM; Table S3). In addition, in the enhanced H08 model, retention areas are modeled such that the retention areas receive runoff in each grid cell; this runoff constitutes a portion of retention pond storage. Therefore, the maximum storage capacity of retention ponds (615 MCM and 935 MCM; Table S3) cannot be utilized for the storage of diverted floodwater. The storage capacity of the retention areas can be increased by changing the maximum water depth based on regional characteristics.

### 4.3 Impact of canal systems on flood control

Initially, the impact of canal systems and retention areas on reducing the annual average, wet season, and dry season discharges of the CPRB was analyzed. Fig. 7 shows the annual, wet season, and dry season discharges in the CPRB for various simulations averaged from 1980 to 2004. The maximum annual average discharge under the NAT simulation was approximately 850 m$^3$/s in the basin (Fig. 7a1), which may lead to devastating impacts in the lower basin, including Bangkok City. The effect of reservoir operation on annual average discharge was negligible (Fig. 7b1). A marked reduction in

discharge, with values ranging between 500 $m^3$/s and 583 $m^3$/s, occurred after enabling irrigation water abstraction (Fig.
7c1). The impact of water diversion on annual average discharge shows that diversion has a great potential for flood control
in the lower CPRB (Fig. 7d1 and e1). In the REG simulation, the annual average discharge of the CPRB was approximately
523 $m^3$/s, a reduction of 10% from the IRG simulation. In contrast, the GEN simulation portrayed a reduction of 28% in
basin annual average discharge, compared with the IRG simulation.

The discharge reduction under various simulations during the wet season was very similar to the annual average flow
reduction pattern except for the DAM simulation. Under the DAM simulation, remarkable discharge reduction was observed
in the Ping, Nan, and Chao Phraya rivers (Fig. 7b2) due to the operation of upstream dam reservoirs of Bhumibol and Sirikit
with an outlet discharge deduction of nearly 15%. The subsequent discharge simulations of IRG, REG, and GEN ones
illustrated nearly 34%, 10%, and 26% reduction in the outlet discharge of the CPRB (Fig. 7c2-e2), which further revealed
the dominance of wet season irrigation water abstraction and canal operations in the annual pattern. During the dry season,
the outlet discharge was nearly 202 $m^3$/s under the NAT simulation (Fig. 7a3), which was enhanced to 551 $m^3$/s after
enabling the reservoir operation (Fig. 7b3). Likewise the wet season, further reductions in discharge was noted in the river
channel due to the irrigation water abstraction and canal operations (Fig. 7c3-e3). Overall, the canal operations significantly
reduced the main channel discharge both annually and intra-annually.


Although water diversion occurred from the Yom River to the Nan River, the impact of this diversion on the annual average
and wet season discharges was negligible. Moreover, the effect of this water diversion in the upper basin was nullified at C.2
station, where the Yom and Nan Rivers join. This finding elucidates the need for (i) water diversion systems to divert
floodwater from the upper to lower basin, (ii) more retention areas to store floodwater, and (iii) other structural and non-
structural measures. Jamrussri et al. (2018) also suggested that new retention areas adjacent to the Yom and the Nan Rivers
are necessary to overcome severe flooding in the upper basin. The effect of water diversion on discharge is more pronounced
in the lower CPRB.

Furthermore, the effect of canal operation on flood control was examined by calculating the number of flooding days (days
during which daily discharge exceeded the channel carrying capacity) at various locations, as shown in Fig. 8. The number
of flooding days are identical for NAT and DAM simulations at Yom River stations because no dams are present in the Yom
basin. The results from the Yom basin showed that the number of flooding days became zero at Y.4 and Y.17 stations in the
REG simulation. Both schemes produced similar results for Y.16 station because of its small channel carrying capacity of
207 $m^3$/s. The reduction of flooding days was relatively small under the GEN scheme in the Yom basin. In the lower CPRB,
the number of flooding days approached zero at C.2 station after the operation of upstream reservoirs; no water diversion
system effects were found. Similarly, the effect of reservoirs was more pronounced at stations in the lower basin, compared
with the effect of diversion canals. Both diversion schemes produced a similar number of flooding days, except at Ayutthaya

(C.35 station). This difference at C.35 station can be explained by the low river channel carrying capacity and high canal carrying capacity at the origins of the Phong Pheng and Bang Ban canals in the REG simulation. Although the number of flooding days was significantly reduced in the REG simulation, the annual average discharge reduction was relatively small because most diverted water was returned to the river channel at some point downstream of the diversion.

Overall, both canal schemes reduced the flood risk, bringing the number of flooding days to nearly zero. Therefore, the incorporation of diversion canals in combination with reservoir operation could mitigate historic floods. However, climate change will aggravate the risk of flooding in the basin (Padiyedath Gopalan et al., 2021). In light of future climate change, new approaches combining structural and non-structural measures must be adopted in the CPRB.

**4.4 Applicability of the generalized canal scheme**

Although the GEN scheme showed some differences from the observations and REG scheme, it was able to identify the general pattern of flow diversion. The combined effect of canal systems in the GEN scheme was identical and comparable with the observations and REG scheme during the wet season, even though the effect of individual canals differs (Fig. 5; Table 4). However, it overestimated the diverted river flow values during the dry season. Considerable overestimation of canal flow was observed mainly for the Makham Thao-Uthong and Chainat-Ayutthaya canals (Fig. 5) during the wet season. This result was obtained because the main purpose of these canals is irrigation water supply, and they have low observed canal carrying capacities (Table S1). However, the GEN scheme considers the primary purpose of all canals to be flood control (because flood control is the primary objective of this study) and fixes the canal carrying capacity at the $Q_{50}$ value, leading to the overestimation of canal flow. This GEN scheme can be improved by differentiating the purpose of each canal in the simulations, similar to the approach in the REG simulation.

The GEN canal scheme was able to produce an annual diversion pattern similar to the annual diversion pattern of the REG scheme (Fig. 6); diverted water comprised approximately 57% of the annual average river discharge in the CPRB. Although this result represents a slightly exaggerated discharge reduction scenario compared with the REG scheme (wherein 49% of annual average river discharge is diverted), it provides insight into the maximum level of discharge reduction that is achievable by maintaining environmental flow levels in the river channel. The GEN scheme showed a greater reduction of basin-wide annual average discharge, which can be attributed to the (i) increased carrying capacity of canals flowing out of the basin (Fig. S3), and (ii) greater retention area for storage of floodwater (Table S3). This finding reveals that the GEN scheme provides more control over the annual average discharge reduction, compared with the REG scheme (Fig. 7). Furthermore, the GEN scheme has a reasonable effect on the flood risk reduction, considerably reducing the number of flooding days (Fig. 8); thus, it can be regarded as the preliminary survey of future global applications using simple input data settings that can be derived from river discharge alone.

## 5 Discussion

To our knowledge, this study is the first attempt to include flood diversion canals and retention areas into GHMs for controlling flood risk. The enhanced H08 model with the REG canal scheme successfully reproduced the observed flood diversion scenario in the study region. The GEN canal scheme exhibited performance comparable with the REG scheme for floodwater diversion. Using existing canal systems, both schemes could divert at least half of the annual average discharge in the CPRB, although some of the diverted discharge gradually returns to the river network. This return flow will increase the subsequent flood risk in downstream areas of the river, although the retention areas will reduce the magnitude of flooding. This finding calls for additional water diversion systems that divert water out of the basin and into nearby basins or the Gulf of Thailand, thereby reducing overall flood risk in the basin (JICA, 2018).

Impact assessment of both canal schemes in terms of river discharge revealed that the number of flooding days could be considerably reduced. Therefore, the enhanced H08 model provides a new tool for assessing future adaptation possibilities of the diversion canal systems. However, several uncertainties and limitations remain to be addressed in future studies. The major uncertainties and limitations of this study can be summarized as follows:

(i)   Pumping stations were constructed in the CPRB to drain the floodwater stored in the retention areas (paddy fields) to the canals immediately after the floods and further operated to drain water from canals to main rivers (JICA, 2013). This pumping process will prepare the paddy fields for cultivation and further storage of water if floods occur within the same year. However, this pumping process was not modeled in this study due to the challenges involved. Instead, the retention storage is assumed to evaporate fully at the end of a normal year and retention storage at the beginning of the following year is set at zero. The evaporation of retention storage at the end of every year is an assumption for simplicity at this stage and further research will be pursued into this area by considering the pumping process in the canal operation scheme.

(ii)  An assumption was also made for the canal operations that the water transfer and retention storage occur without any loss and delay. The evaporation and conveyance loss while canal water transfer was assumed to be trivial and the delay in delivery at the canal destination is not taken into account. In addition, a part of the retention storage might be lost through percolation. However, due to limitations in the availability of such data, the percolation loss was assumed to be negligible in the highly saturated paddy fields at the time of floods.

(iii) The canal water is assumed to be supplied to the grid cells through which the canal passes as well as to the immediate lateral neighbouring grid cells of the canal for meeting the irrigation demand. Therefore, the spatial resolution of the simulation will affect the results. In this study, for simplicity, only 10% of diverted water is supplied to each of the nearby grid cells because our primary concern was flood control. Therefore, this fraction of 'supply to near grids' should change if the simulation is performed on a finer/coarser resolution. One alternative way to overcome this issue is that the 'supply to near grids' can be finalized based on the water demand in each of the grid cells through which the

canal passes as well the in the neighbouring grid cells. In such instances, it can be confirmed that the supplied water will be completely utilized.

(iv) The assumption of rainfed croplands as retention areas will affect the crop yield because the cropland is not available for cultivation until the beginning of the subsequent year due to the storage of floodwater. This is one of the limitations this study currently poses and that will be addressed in our future studies by considering the actual pumping process in the canal operation scheme. However, this is not a universal assumption for future global applications. The most important land use for potential retention areas is the low-lying areas along rivers (floodplains) and canals. Historically,

such lowland is used for paddy cultivation in warm Asian countries. Being paddy is not the required condition for retention areas. In addition, although the lakes/ponds could be partially filled with water during the wet season, they can also be used as retention areas based on available free space. The geographic locations of such possible retention areas along with their depth and areal extents available for storage of floodwater specific to each area can be extracted from remotely sensed data such as digital elevation models (e.g., MERIT DEM), satellite imageries

(MODIS/LANDSAT), radar altimetry, as well as from literature although it is strenuous.

(v) Another limitation mainly arises from the assumptions made for river channel carrying capacity, canal carrying capacity, and minimum flow of diversion in GEN canal scheme simulations.

Acquisition of observations and accurate representation of these variables in GHMs remain challenging tasks. However, this

study represents a first step toward developing and incorporating a diversion scheme into the H08 GHM. The robustness of this scheme can be improved by adapting and applying it to different contexts and regions, accounting for the characteristics of each river network while maintaining the basic principles unaltered.

**6 Conclusions**

Assessing the impacts of flood diversion canals on flood risk reduction worldwide remains challenging. To overcome this

issue, a flood diversion canal operation scheme was developed for the CPRB, a complex river network in Thailand with several natural and man-made diversion canals. The developed scheme was carefully designed and implemented into the H08 GHM for future floodwater management in complex river networks worldwide. A generalized canal scheme was also introduced with simple input data settings that can be derived from river discharge alone for application to data-scarce regions and periods. The major conclusions from this study can be summarized as follows:

1. A regionalized water diversion scheme was developed for the CPRB and incorporated into the H08 GHM to evaluate the effects of floodwater diversion on water-donating and its downstream areas, with the primary purpose of flood management. The validation results show that the diversion scheme can effectively simulate the observed flood diversion scenario in the CPRB. The generalized water diversion scheme with simple input data settings also exhibited comparable performance, especially during the wet season.

2. The major share of the diverted water gradually returns to the destination points under both canal schemes. This indicates that although flood risk is reduced at the point of diversion, some risk remains in the destination area. If retention areas were able to hold more water, flood damage to residential and commercial areas of the lower CPRB could be reduced. This change can be achieved by (i) considering retention areas in irrigated and non-irrigated croplands simultaneously, (ii) assuming that the water from retention areas can be drained into canals and rivers after the floods using pumps; this makes the retention areas available for future floods in the same year, and (iii) increasing the depth of water stored in the retention areas (a depth of one m was assumed in this study).

3. Water diversion led to a marked reduction in the annual average discharge of the CPRB, which was much greater under the generalized canal scheme than the regionalized scheme. Furthermore, both canal schemes reduced the number of flooding days to nearly zero at most of the gauging stations considered. The overall simulation results indicate that currently implemented canal schemes have the potential to reduce flood risk in the upper and lower CPRB, where many industrial and residential areas are located. However, their abilities to overcome severe flooding should be further evaluated in the context of climate change.

Therefore, we emphasize that the regionalized canal scheme described herein was successfully applied to the CPRB, whereas the generalized scheme requires further validation to evaluate its applicability in other regions worldwide with slight modification. The water diversion rules implemented in the H08 model (during wet and dry seasons) can be easily adapted to different GHMs and should be examined for their applicability. In future research, we will expand upon these schemes for future applications by considering various adaptation scenarios.

**Code availability**

The source code and operation manual of H08 model is freely available from the following website: https://h08.nies.go.jp/h08/index.html. The code for the diversion canal systems used in this study are available from the authors upon reasonable request.

**Data availability**

The meteorological data used in study is available from IMPAC-T project website (http://impact.eng.ku.ac.th/cc/). The remaining data that support the findings of this study are available from the authors upon reasonable request.

**Author contribution**

SPG and NH conceived the idea. AC and TS acquired the data. SPG, NH, and SN analyzed and interpreted the data. SPG developed the model code and performed the simulations. SPG prepared the manuscript with contributions from all co-authors. NH, AC, TS, and SN reviewed and edited the manuscript.

**Competing interests**

The authors declare that they have no conflict of interest.

**Acknowledgements**

This study was carried out as a part of the research project entitled "Advancing Co-Design of Integrated Strategies with Adaptation to Climate Change in Thailand (ADAP-T)" supported by the Science and Technology Research Partnership for Sustainable Development (SATREPS) program of the Japan Science and Technology Agency (JST) and the Japan International Cooperation Agency (JICA).

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

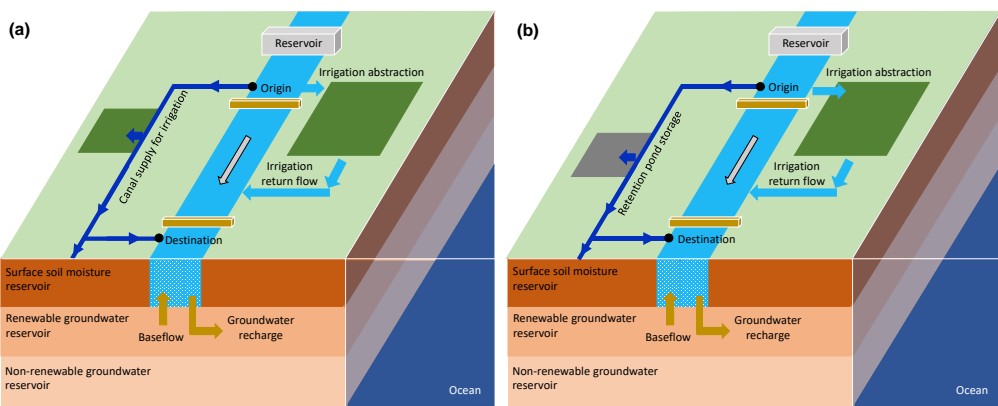

**Figure 1: Schematic diagram of the canal diversion scheme of the enhanced H08 model during the (a) dry season and (b) wet season. Blue, green, orange, and grey symbols denote water, land, underground reservoirs, and retention areas, respectively.**

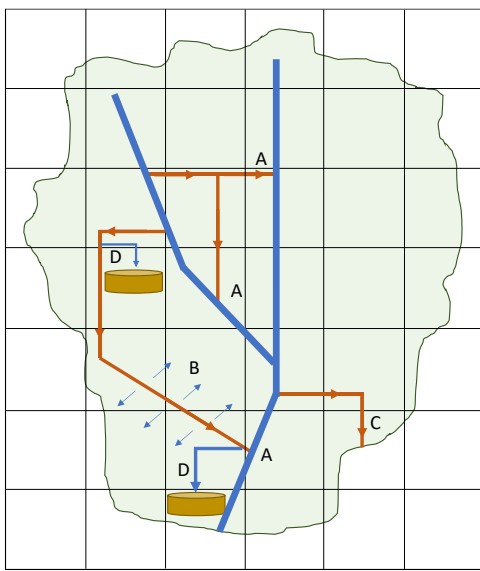

A - Returned discharge    C - Flow out of the basin

B - Supply to near grids    D - Retention storage

**Figure 2: Schematic diagram of the four major components of the diverted canal flow. The green area denotes the river basin.**


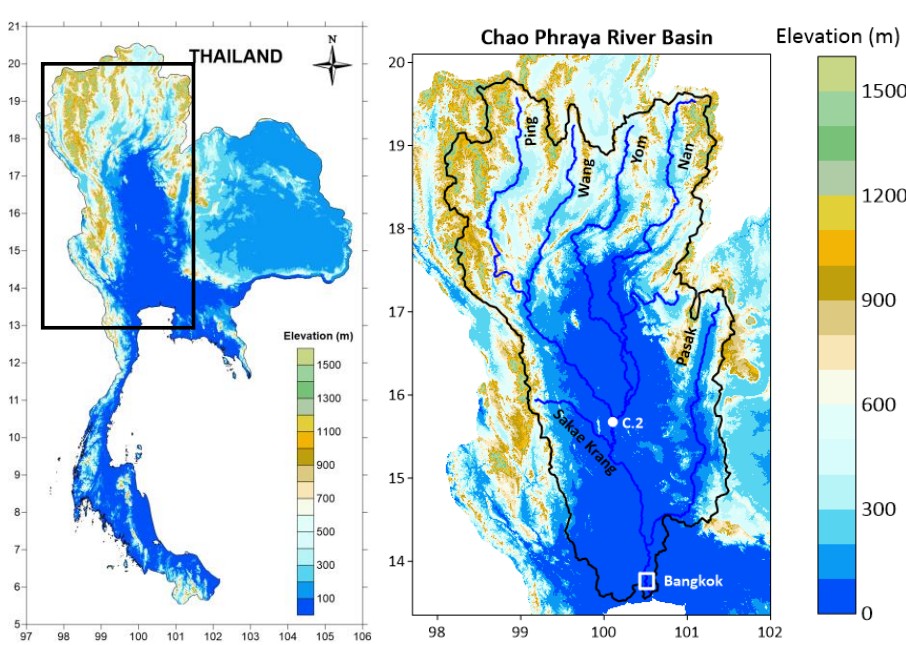

**Figure 3: Left: Topographic map of Thailand in which the rectangle shows the location of the Chao Phraya River basin (CPRB). Right: Topographic map of the CPRB with the major tributaries and the locations of the Nakhon Sawan (C.2 station) and Bangkok City.**

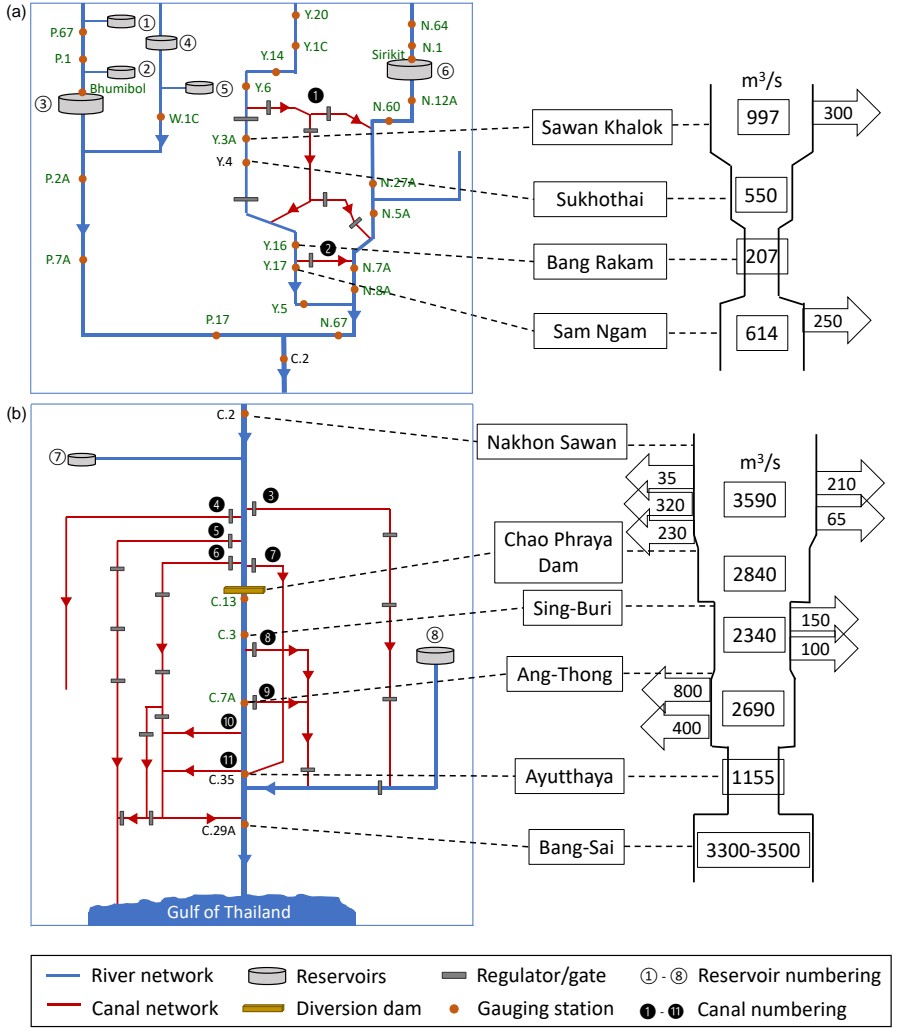

**Figure 4: Left: Schematic diagram of different existing water infrastructures in the (a) upper CPRB and (b) lower CPRB. Right: Observed river channel carrying capacity shown inside the river schematic and the observed canal carrying capacity shown in the left and right arrows for various locations within the CPRB. Please refer to Table 1 for the canal numbering and Table S2 for the reservoir numbering. The green colour gauging stations represent validation locations.**

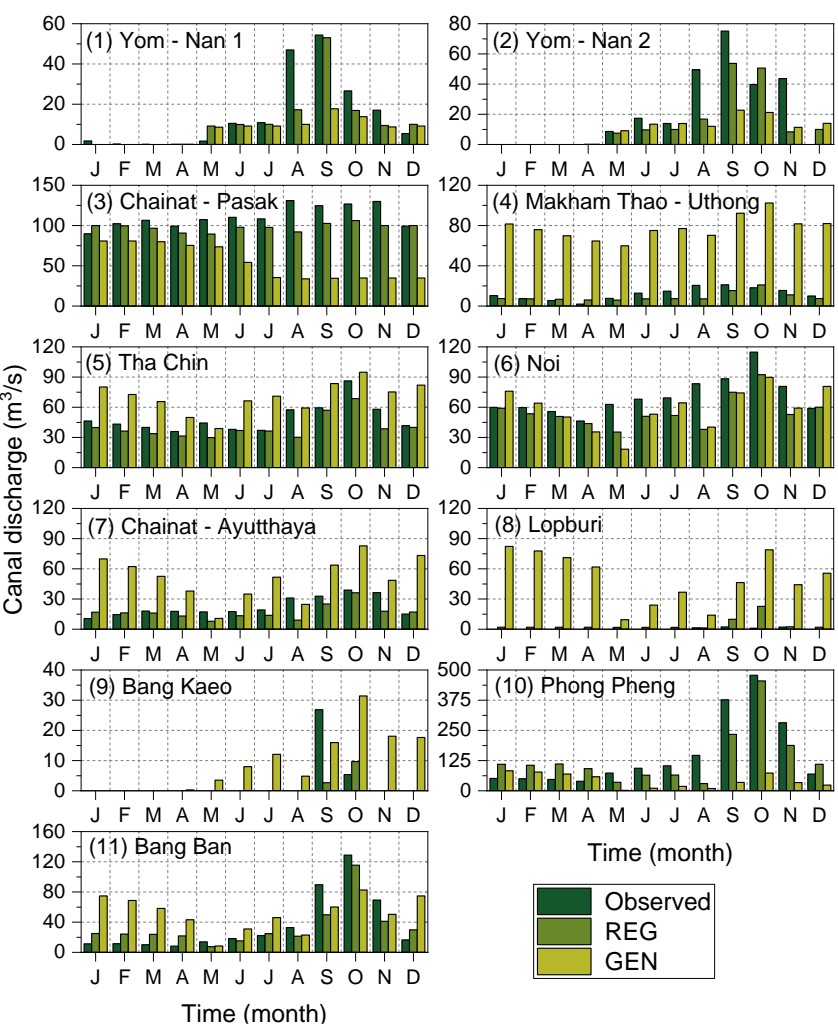

Figure 5: Simulated monthly mean canal discharge (REG and GEN) at eleven canal origins (in the order shown in Figure 4) compared with the observed flow.

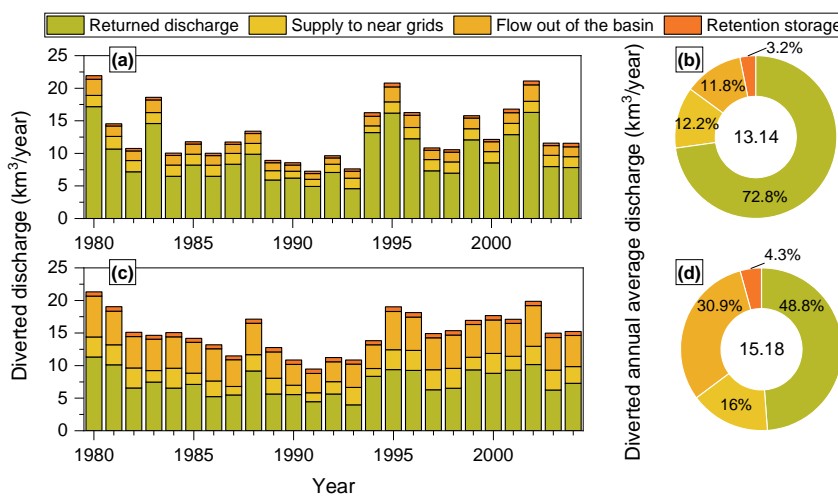

**Figure 6: Amount of flow transferred from the canal systems to each of the components (from 1980-2004) and their percentage contribution to the annual canal flow averaged over 1980 to 2004 under the regionalized scheme (a and b) and generalized scheme (c and d).**

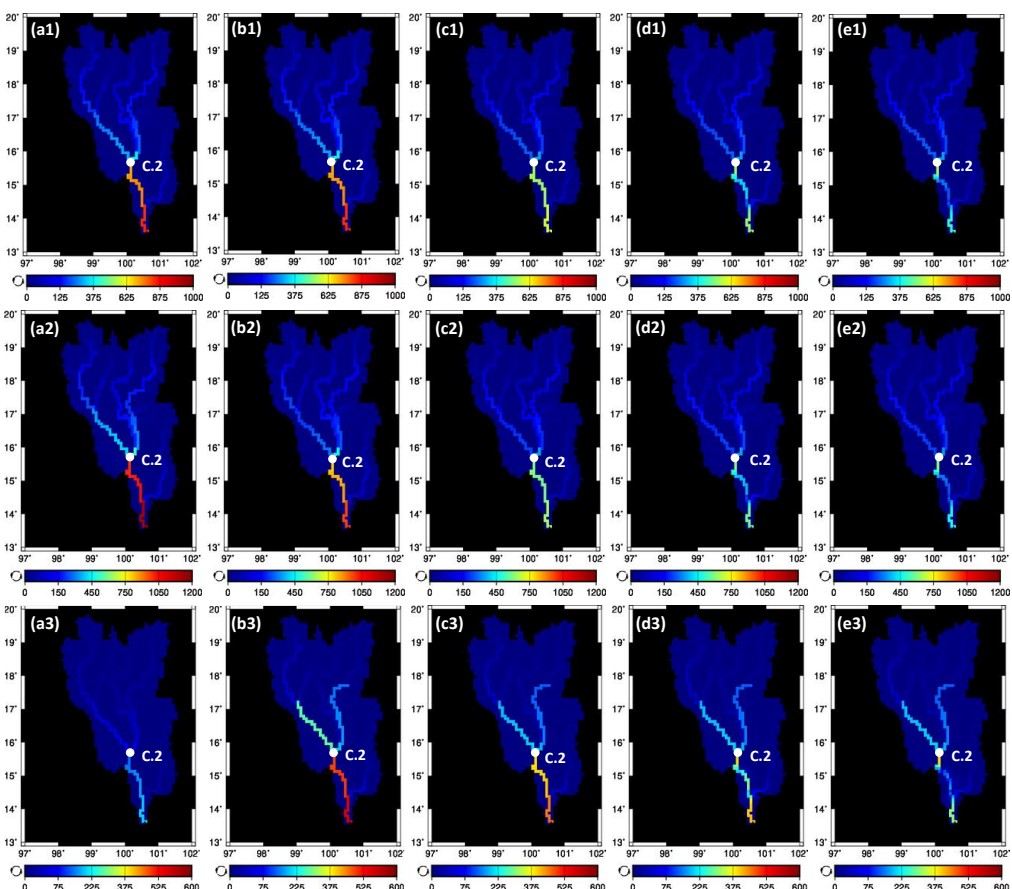

**Figure 7: Effect of canal systems on annual average (Top panel), wet season (middle panel), and dry season (bottom panel) discharges in the CPRB for the NAT (a1-a3), DAM (b1-b3), IRG (c1-c3), REG (d1-d3), and GEN (e1-e3) simulations.**

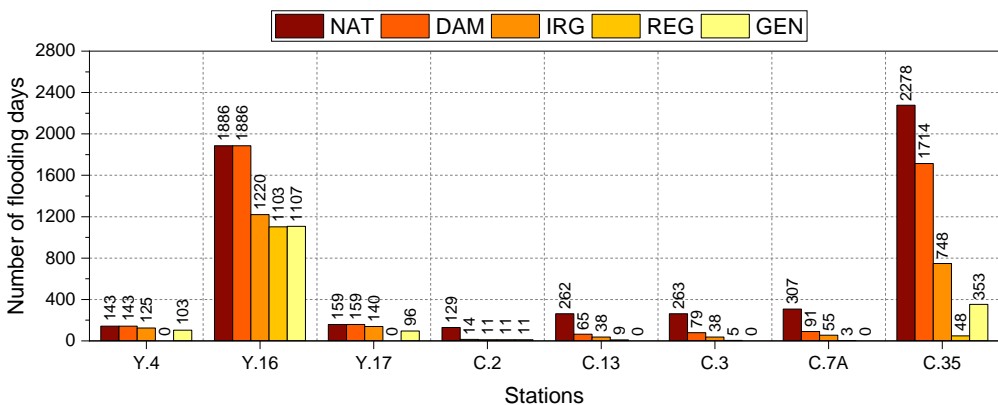

**Figure 8: The number of days the daily discharge exceeded the channel carrying capacity from 1980-2004 (25 years) at various locations (refer to Fig. 4 for the channel carrying capacity of each station) under different simulations.**

**Table 1. Origin and destination points of the canal systems in the CPRB with their regulating structures and purpose.**

| No. | Canal system | Origin | Destination | Regulating structure | Purpose |
|---|---|---|---|---|---|
| 1 | Yom – Nan 1 | Yom | Nan, and downstream of Yom | Ban Hat Saphan Chan barrage | Flood control |
| 2 | Yom – Nan 2 | Yom | Nan | DR. 2.8 | Flood control |
| 3 | Chainat – Pasak | Chao Phraya | Pasak | Manorom barrage | Irrigation supply and flood control |
| 4 | Makhamthao – Uthong | Chao Phraya | Out of the basin | Makamthao-Uthong barrage | Irrigation supply and flood control |
| 5 | Tha chin | Chao Phraya | Gulf of Thailand | Pholthep barrage | Irrigation supply and flood control |
| 6 | Noi | Chao Phraya | Downstream of Chao Phraya | Borommathat barrage | Irrigation supply and flood control |
| 7 | Chainat – Ayutthaya | Chao Phraya | Downstream of Chao Phraya | Maha Raj barrage | Irrigation supply and flood control |
| 8 | Lopburi | Chao Phraya | Pasak | Lopburi barrage | Flood control |
| 9 | Bang Kaeo | Chao Phraya | Lopburi | Bang Kaeo barrage | Flood control |
| 10 | Phong Pheng | Chao Phraya | Noi | Gravity transfer | Irrigation supply and flood control |
| 11 | Bang Ban | Chao Phraya | Noi | Gravity transfer | Irrigation supply and flood control |


**Table 2. The differentiation of regionalized and generalized canal schemes in terms of the input data (JICA, 2013; Tamada et al., 2013)**

| Input data | Regionalized canal scheme | Generalized canal scheme |
|---|---|---|
| Geographic location of canal origin and destination | Observed | Observed |
| Location and areal fraction of retention areas | Areal extent of rainfed agriculture + Observed | Areal extent of rainfed agriculture |
| River channel carrying capacity ($Q_{rivcap}$) | Adjusted | $Q_5$ value |
| Canal carrying capacity ($Q_{cancap}$) | Adjusted | $Q_{50}$ value |
| Minimum flow diversion ($Q_{min}$) | Adjusted | $Q_{90}$ value |
| Days of diversion in a year | Observed | Observed |


**Table 3. The different simulations carried out in this study.**

| Simulations | Reservoir operation | Irrigation water abstraction | Regionalized canal scheme | Generalized canal scheme |
|---|---|---|---|---|
| Naturalized (NAT) | ☒ | ☒ | ☒ | ☒ |
| Dam (DAM) | ☑ | ☒ | ☒ | ☒ |
| Irrigated (IRG) | ☑ | ☑ | ☒ | ☒ |
| Regionalized (REG) | ☑ | ☑ | ☑ | ☒ |
| Generalized (GEN) | ☑ | ☑ | ☒ | ☑ |



**Table 4. The observed and simulated fractions of diverted river flow to the canals under regionalized (REG) and generalized (GEN) canal schemes during the wet and dry seasons.**

| Canal system | Reference gauging station for canal water diversion | Dry season (%) | | | Wet season (%) | | |
|---|---|---|---|---|---|---|---|
| | | Observed | REG | GEN | Observed | REG | GEN |
| 1 | Y.14 | 0 | 0 | 0 | 19.91 | 20.75 | 13.22 |
| 2 | Y.16 | 0 | 0 | 0 | 15.33 | 17.70 | 12.00 |
| 3 | C.2 | 26.18 | 25.12 | 20.58 | 14.32 | 16.69 | 7.18 |
| 4 | C.2 | 1.66 | 1.78 | 18.96 | 1.84 | 1.75 | 13.67 |
| 5 | C.2 | 10.89 | 9.1 | 17.41 | 6.44 | 7.17 | 12.18 |
| 6 | C.2 | 14.60 | 13.44 | 14.67 | 9.56 | 9.69 | 10.23 |
| 7 | C.2 | 4.00 | 4.04 | 14.45 | 3.17 | 2.98 | 8.33 |
| 8 | C.3 | 0 | 0.71 | 81.70 | 0.13 | 1.52 | 15.42 |
| 9 | C.7A | 0 | 0 | 0 | 0.83 | 0.43 | 5.89 |
| 10 | C.7A | 36.45 | 34.60 | 43.36 | 41.97 | 41.48 | 10.86 |
| 11 | C.7A | 8.02 | 7.84 | 36.93 | 10.12 | 10.73 | 19.88 |
| Average | | 9.25 | 8.78 | 22.55 | 11.24 | 11.90 | 11.71 |




# Supplement of Inclusion of flood diversion canal operation in the H08 hydrological model with a case study from the Chao Phraya River Basin: Model development and validation

Saritha Padiyedath Gopalan[1], Adisorn Champathong[2], Thada Sukhapunnaphan[2], Shinichiro Nakamura[3], Naota Hanasaki[1]

[1]Centre for Climate Change Adaptation, National Institute for Environmental Studies (NIES), 16-2 Onogawa, Tsukuba, Ibaraki 305-8506, Japan
[2]Royal Irrigation Department, 811 Samsen Road, Dusit, Bangkok, Thailand
[3]Royal Department of Civil Engineering, Graduate School of Engineering, Nagoya University, Building No. 9, Furo-cho, Chikusa-ku, Nagoya 464-8603, Japan

*Correspondence to*: Saritha Padiyedath Gopalan (pgsaritha@nies.go.jp; charu666@gmail.com)

**S1 Schematic of the canal diversion scheme**

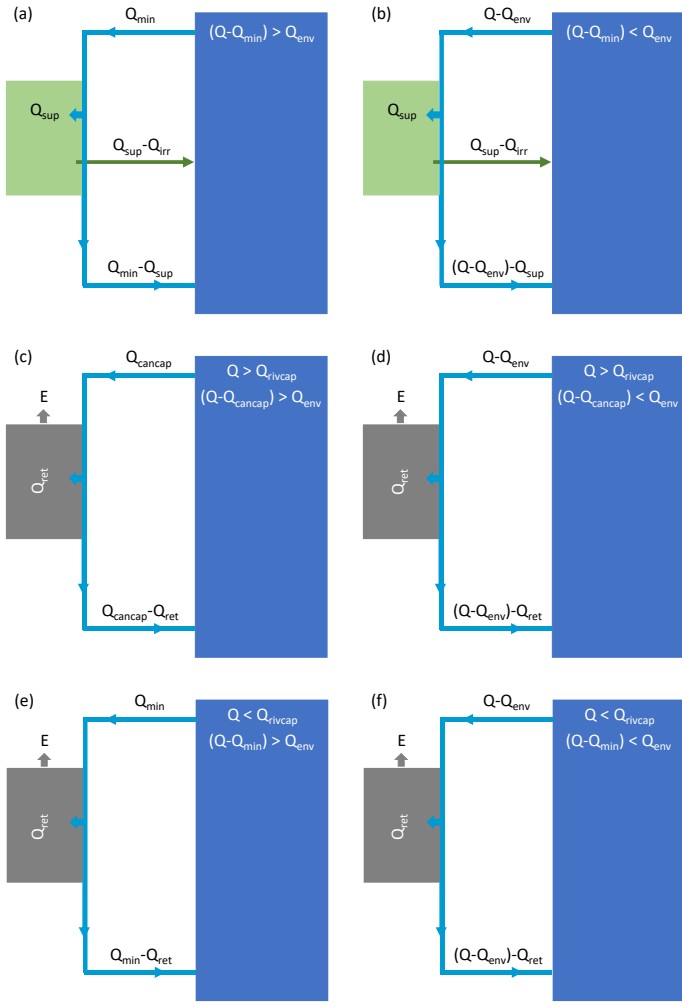


**Figure S1: Schematic diagram of the canal diversion scheme during the (a-b) low flow in dry season, (c-d) flood flow in wet season, and (e-f) non-flood flow in wet season. Blue, green, and grey symbols denote river, agricultural land, and retention areas, respectively. Blue and green arrows represent the canal flow and return flow from agricultural land, respectively.**

**S2 Water infrastructures in the CPRB**

**S2.1 Canal system**

Table S1 shows the observed river channel carrying capacity, canal carrying capacity, and minimum flow of diversion at the origins of the eleven canal systems. Table S1 also shows the simulated values under the regionalized and generalized schemes as explained in section S3.

**Table S1. Observed (OBS) and simulated (REG for regionalized and GEN for generalized simulations) values of river channel carrying capacity at the canal origin, canal carrying capacity, and the minimum flow diversion of each canal for the CPRB.**

| No. | Canal system | River channel carrying capacity (m³/s) | | | Canal carrying capacity (m³/s) | | | Minimum flow diversion (m³/s) | | |
|-----|--------------|------|------|------|------|------|------|------|------|------|
| | | OBS | REG | GEN | OBS | REG | GEN | OBS | REG | GEN |
| 1 | Yom – Nan 1 | 850 | 400 | 284.3 | 300 | 300 | 42.0 | 10 | 10 | 9.2 |
| 2 | Yom – Nan 2 | 600 | 400 | 414.1 | 250 | 250 | 62.9 | 10 | 10 | 14.0 |
| 3 | Chainat – Pasak | 2000 | 2500 | 2356.2 | 210 | 210 | 362.9 | 100 | 100 | 80.9 |
| 4 | Makham Thao – Uthong | 2000 | 1000 | 2379.6 | 35 | 35 | 365.8 | 6 | 7.5 | 82.0 |
| 5 | Tha chin | 2000 | 2000 | 2382.4 | 320 | 320 | 366.1 | 40 | 40 | 82.0 |
| 6 | Noi | 2000 | 1500 | 2384.0 | 230 | 230 | 367.4 | 55 | 60 | 82.6 |
| 7 | Chainat – Ayutthaya | 2000 | 1000 | 2393.1 | 65 | 65 | 366.8 | 15 | 17 | 82.1 |
| 8 | Lopburi | 2900 | 1500 | 2431.1 | 150 | 150 | 376.5 | 0 | 2 | 85.4 |
| 9 | Bang Kaeo | 2800 | 1500 | 2449.0 | 100 | 100 | 380.6 | 0 | 0 | 87.1 |
| 10 | Phong Pheng | 1000 | 900 | 2456.7 | 800 | 1000 | 381.2 | 45 | 110 | 87.4 |
| 11 | Bang Ban | 1000 | 650 | 2457.6 | 400 | 400 | 383.3 | 10 | 25 | 87.8 |

**S2.2 Reservoirs**

Details of the eight multipurpose reservoirs in the CPRB are provided in Table S2. Operation data for all reservoirs were obtained from the Electricity Generating Authority of Thailand (EGAT) and the Royal Irrigation Department (RID), Thailand. For each of the reservoirs, releases during the wet and dry seasons were calculated based on the long-term mean of observed reservoir release data. These long-term mean release values were bias-corrected with respect to simulated inflow because of the difference between observed and simulated inflows into the reservoirs. These bias-corrected releases were then adjusted with reference to reservoir storage targets or limits, which are set based on the upper and lower storage guide curves; these curves critically affect the simulated volume of water stored in the reservoirs. Detailed information regarding the reservoirs and their operation in the CPRB is available from Padiyedath Gopalan et al. (2021) and Mateo et al. (2014).

**Table S2. The details of the existing reservoirs in the CPRB (Lehner et al., 2011).**

| No. | Reservoir | Year of construction | Storage capacity (MCM) | Catchment area (km²) | Main purposes |
|---|---|---|---|---|---|
| 1 | Mae Ngat | 1985 | 265.0 | 1281 | Irrigation<br>Water supply |
| 2 | Mae Kuang | 1991 | 263.0 | 558 | Irrigation<br>Water supply |
| 3 | Bhumibol | 1964 | 13462.0 | 26400 | Irrigation<br>Flood control<br>Water supply<br>Hydroelectricity |
| 4 | Kiew Lom | 1972 | 112.0 | 2747 | Irrigation<br>Water supply |
| 5 | Mae Chang | 1983 | 108.6 | 290 | Water supply<br>Hydroelectricity |
| 6 | Sirikit | 1974 | 9510.0 | 13130 | Irrigation<br>Flood control<br>Water supply<br>Hydroelectricity |
| 7 | Thap Salao | 1988 | 160.0 | 531 | Irrigation<br>Water supply |
| 8 | Pasak | 1999 | 960.0 | 12970 | Irrigation<br>Flood control<br>Water supply |

**S3 Inclusion of canal systems and retention areas in the H08 model**

Eleven canal systems were digitized into the H08 model under the regionalized and generalized schemes, as shown in Fig. S2(b) and (c), respectively. The Chainat-Pasak canal, on the left bank of the Chao Phraya River in the lower CPRB, flows through the Pasak River before emptying into the Gulf of Thailand (Fig. S2a). However, in this study, the Pasak River was regarded as the destination point of the Chainat-Pasak canal because further downstream data were unavailable (Fig. S2b and c).


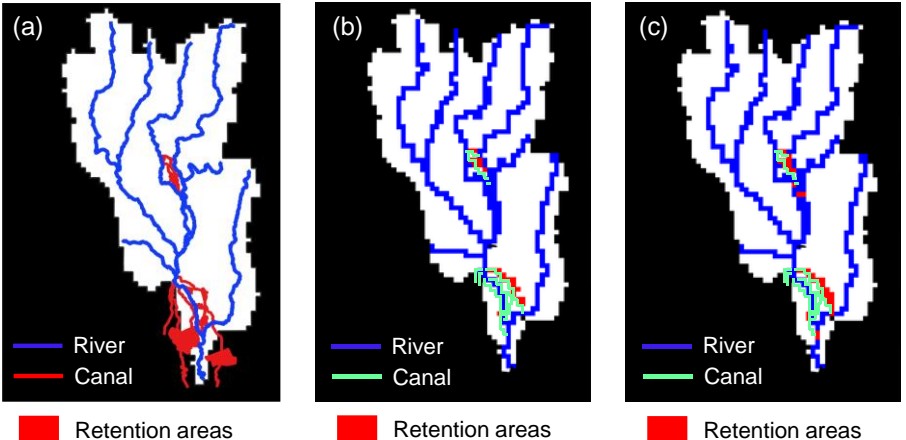

Figure S2: Visual comparison of the canal networks and retentions areas of the CPRB in the (a) observed, (b) regionalized, and (c) generalized canal schemes.

The observed and modeled (regionalized and generalized) areas of the retention ponds associated with each of the canal systems are provided in Table S3. No retention areas were associated with the Yom-Nan 2 and Bang Kaeo canals; retention areas of the Makham Thao-Uthong and Tha Chin canals were excluded from analysis because both the canal and the retention areas lie outside the basin, as shown in Fig. S2(a). The modeled retention areas were smaller in area than the observations for most of the canal systems because there were few rainfed croplands near those networks. The total area of
retention ponds was approximately 1702 km$^2$, whereas the retention areas obtained under the regionalized and generalized schemes were 615 km$^2$ (approximately one-third of the observed area) and 935 km$^2$ (approximately half of the observed area), respectively. The small retention areas simulated in the regionalized scheme can be attributed to the refinement of data conducted to match the data provided by the RID.

**Table S3. Observed and modelled (regionalized and generalized) areas of the retention areas associated with each canal system.**

| Basin | Canal system | Observed area (km$^2$) | Regionalized area (km$^2$) | Generalized area (km$^2$) |
|---|---|---|---|---|
| Upper CPRB | Yom – Nan 1 | 424.00 | 276.02 | 365.96 |
| | Yom – Nan 2 | 0 | 0 | 54.53 |
| Lower CPRB | Chainat – Pasak | 250.69 | 223.07 | 294.14 |
| | Noi River | 780.21 | 0 | 14.23 |
| | Chainat – Ayutthaya | 27.20 | 26.40 | 26.40 |
| | Lopburi | 132.8 | 89.64 | 154.16 |
| | Bang Kaeo | 0 | 0 | 0 |
| | Phong Pheng | 33.37 | 0 | 0 |
| | Bang Ban | 53.52 | 0 | 25.87 |
| Total | | 1701.79 | 615.13 | 935.29 |

Before conducting canal simulations, values of variables such as the river channel carrying capacity, canal carrying capacity, and the minimum flow of diversion were set for both canal schemes. In the regionalized scheme, an adjusted version of the observed values of these variables was used for the H08 model because the simulated discharge was slightly lower than the

observed discharge at various diversion locations. In the generalized scheme, $Q_5$, $Q_{50}$, and $Q_{90}$ values were used to represent river channel carrying capacity, canal carrying capacity, and minimum flow of diversion, respectively.

Fig. S3 is a scatter plot comparing the values of the river channel carrying capacity, canal carrying capacity, and minimum flow of diversion for the regionalized (top panel) and generalized (bottom panel) canal schemes to the observed values. The

regionalized estimates exhibited slight variation from the observations because of the adjustment noted above. Under the generalized scheme, the river carrying capacity values were underestimated at the origin of flood diversion canals while overestimated for multi-purpose channels. Although these values exhibited slight variations, they were comparable with the observations, except in two canal systems (Phong Pheng and Bang Ban; Table S1). The low observed river carrying capacity at the origins of these two canal systems is to achieve a maximum discharge reduction at Ayutthaya (C.35 station), where the

channel carrying capacity is small. The canal carrying capacity was almost close for most of the canals. Likewise, the minimum flow diversion values for many of the canal systems were similar. This is because of the very small inflow contributions into the lower Chao Phraya River. Their values exhibited deviations with respect to the observations. Most of these deviations were in values for multi-purpose canals, because the primary purpose of all canals under the generalized scheme is flood control. These values were subsequently employed for the canal simulations.

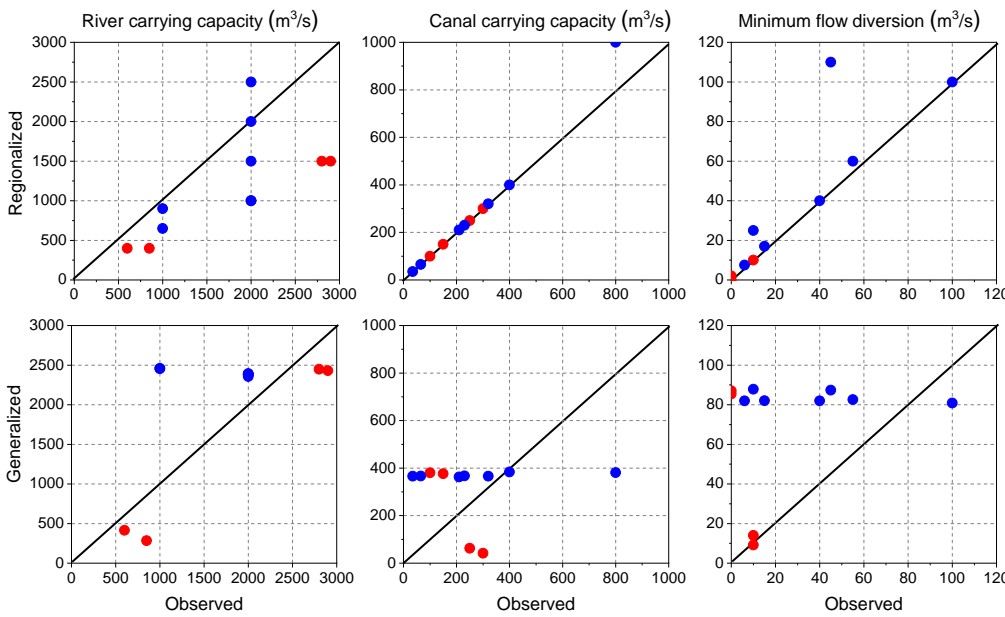

**Figure S3: The scatter plot of the observed values of the river carrying capacity, canal carrying capacity, and the minimum flow of diversion against the values of the regionalized and generalized canal schemes. The red and blue circles represent the flood and multi-purpose canal systems, respectively.**

### S4 H08 model calibration and validation

#### S4.1 Naturalized discharge simulation

The H08 model was recalibrated at Nakhon Sawan for estimation of naturalized discharge (NAT) because new groundwater components were incorporated into the H08 model (Hanasaki et al., 2018). This recalibration was performed by keeping the remaining settings identical to the settings described by Padiyedath Gopalan et al. (2021). The calibrated parameters of the land surface hydrology module for the CPRB are shown in Table S4, along with the corresponding global parameters. Using these calibrated parameters, the NAT discharge was simulated by enabling the land surface hydrology and river routing modules. These modules do not include the effect of water infrastructures and thereby simulate the NAT discharge. Further, the observed naturalized discharge at Nakhon Sawan was reconstructed by removing the effect of two major dam reservoirs (Bhumibol and Sirikit) operating upstream of the station. This was performed by adding the water stored in the two dam reservoirs with the observed discharge at Nakhon Sawan (Mateo et al., 2014). The transformation of observed discharge into

the observed naturalized discharge and the associated uncertainties are described in detail by Champathong et al. (2020). The estimation of the observed naturalized discharge at Nakhon Sawan was carried out using the following equation:

$$Q_{Nat} = Q_{Obs} + [I + P - R - S]_{Bhumibol} + [I + P - R - S]_{Sirikit} \tag{S1}$$

where $Q_{Nat}$ is the observed naturalized discharge, $Q_{Obs}$ is the observed discharge, $I$ is the reservoir inflow, $P$ is the water pumped into the reservoir, $R$ is the reservoir release, and $S$ is the water released through the spillway. Further, the computed observed naturalized discharge was compared with the simulated NAT discharge from the H08 model to examine the hydrograph reproducibility. Naturalized discharge was adequately reproduced at Nakhon Sawan, with daily and monthly Nash-Sutcliffe efficiency (NSE) values of 75.18% and 86.07%, respectively. Furthermore, the observed and simulated annual average river discharges were 701 and 692 $m^3$/s, respectively; these differed by only 1.3%.

In addition, model validation was conducted at 28 stations in the CPRB (Fig. 4) using NSE as the evaluation criterion. The minimum and maximum daily NSE values were 31% and 87%, respectively, with a mean value of approximately 49%. Similarly, the monthly NSE values ranged from 33% to 90%, with a mean value of approximately 67%. Overall, the performance of the H08 model was very good at four stations (monthly NSE values of 75−100%), good at thirteen stations (monthly NSE values of 65−75%), and satisfactory at eight stations (monthly NSE values of 50−65%), based on monthly NSE values (Moriasi et al., 2007). However, unsatisfactory performance was observed at three stations located in the far upstream reaches of the river networks; these monthly NSE values were below 50%.

**Table S4. Calibrated parameters of land surface hydrology module at Nakhon Sawan.**

| Parameters | Global setup | Regional setup |
|---|---|---|
| Soil depth (m) | 1.00 | 2.50 |
| Bulk transfer coefficient | 0.003 | 0.013 |
| Time constant (day) | 100 | 70 |
| Shape parameter | 2.00 | 2.30 |
| Groundwater depth (m) | 1.00 | 0.50 |
| Groundwater yield | 0.30 | 0.10 |
| Groundwater time constant (day) | 2.00 | 4.00 |
| Groundwater shape parameter | 100 | 50 |

**S4.2 Dam discharge simulation**

The ability of the H08 model to explicitly reproduce the observed discharge hydrograph at Nakhon Sawan, using the recalibrated parameters, was evaluated by enabling the reservoir operation module of the H08 model in addition to the land

surface hydrology and river routing modules. This will facilitate the comparison of the observed discharge with the simulated dam discharge (DAM) under the assumption that the DAM discharge could act as a proxy for the observed discharge although precisely not the case because the DAM discharge simulation does not include water abstraction for irrigation. Still, this comparison was made to evaluate the performance of the included reservoir operations in the model. Reservoir operation rules were set in accordance with the operation rules described by Padiyedath Gopalan et al. (2021), based on the upper storage guide curves of historical reservoir operation. The model exhibited good performance, with daily and monthly NSE values of 75.61% and 80.75%, respectively. Moreover, the observed and simulated annual average river discharges were 673 and 686 $m^3$/s, respectively; these differed by only 1.9%.

## S4.3 Irrigated discharge simulation

In Asian countries, canal systems serve as floodways during the wet season and water supply channels during the dry season. Hence, estimating irrigation water demand is crucially important in running the simulation over a year. In the H08 model, the crop growth module estimates the crop calendar and associated crop yields. In the coupled model, irrigation water demand and streamflow were utilized to estimate water withdrawal. Therefore, in this section, we validated the simulated crop calendar, crop yields, and irrigation water withdrawal through the comparison of the results obtained here with previous reports and the expert opinions of RID officials in Thailand.

### S4.3.1 Crop calendar and yield

Initially, we simulated the crop calendar of different crops using the stand-alone crop growth module of the H08 model. For calculation of the crop calendar, we multi-averaged the variables (air temperature, shortwave downward radiation, evapotranspiration, and potential evapotranspiration) that were used to compute the crop calendar from 1980 to 2004. Then by utilizing these multi-year averaged variables, we estimated a single crop calendar for each of the crops in CPRB. Then, we validated the crop calendar of three major crops in Thailand by comparing the simulated crop calendar with the calendar reported in the agricultural handbook of the World Agricultural Outlook Board, United States Department of Agriculture (WAOB-USDA, 1994), which provides planting and harvesting dates for major crops in countries worldwide. The three selected crops are rice (first and second crop), maize, and sugarcane, as shown in Fig. S4.

Rice is the major crop grown in Thailand, with two main cropping seasons known as the first and second rice crops (Titapiwatanakun, 2012). According to WAOB-USDA (1994), the first rice crop is grown mainly from May to August in most of Thailand, and its harvesting period is from October to January of the following year (Fig. S4). The second rice crop is irrigated and grows in the dry season from January to early March, with harvests from May to June (WAOB-USDA, 1994). Another report by Titapiwatanakun (2012) notes that the first and second rice crops are grown from May to October and November to April, respectively, providing a wide cropping calendar with a span of six months related to regional differences in cropping schedules. Overall, by combining the reports of WAOB-USDA (1994) and Titapiwatanakun (2012),

the planting and harvesting dates were adequately reproduced in the simulations for the first and second rice crops, although

they exhibited a lag of approximately one month compared with the WAOB-USDA report, as shown in Fig. S4.

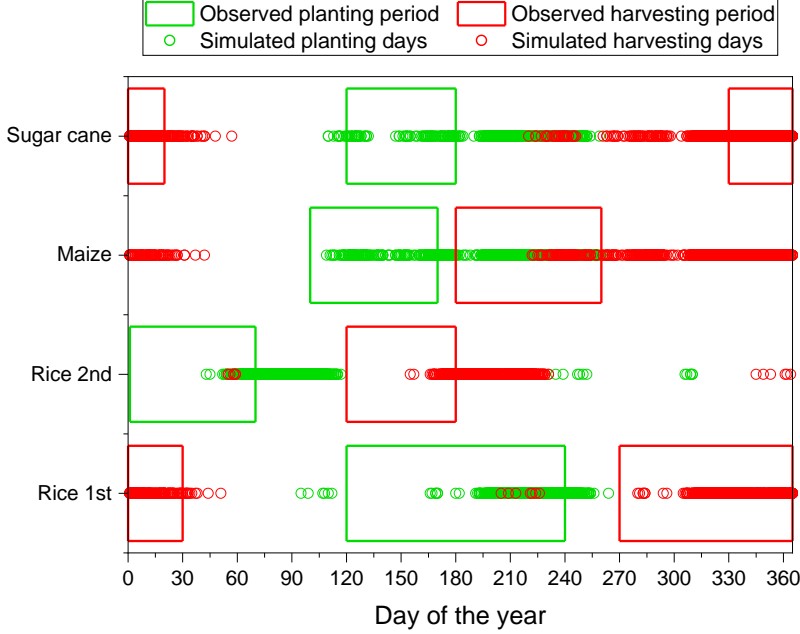

**Figure S4: Observed and simulated planting and harvesting dates for rice (first and second crop), maize, and sugarcane. Each circle represents each grid cell in the H08 model domain.**

For maize, the simulated planting dates were well reproduced. Although early April to late June is the most suitable planting period for maize, it may be planted from March to September based on soil moisture availability (Senanarong, 1968), which was adequately reflected in the simulations. The simulated planting dates aligned with the observations, but they occurred approximately one month later in some regions. Similar to the planting dates, the harvest dates showed a wide range of up to five months. The simulated planting and harvesting dates were fairly captured for sugarcane in most areas. Some regions

showed differences from the observations, reflecting variations in regional conditions. In general, the planting and harvesting dates of major crops in the CPRB closely agreed with the WAOB-USDA data. The exceptions to this tendency included late estimation of both planting and harvest dates by nearly one month, as well as cropping periods longer than the observed data.

Using the simulated crop calendar, we estimated the potential yields of the three major crops and compared the results with

yields in the WAOB-USDA report (Fig. S5). The simulated annual average yields of rice (first crop) and maize from 1980 to 2004 were 4–6 t/ha and 8–10 t/ha, respectively. The simulated yields of these crops were high, compared with observed yields of 1.94 t/ha (rice) and 2.76 t/ha (maize). These differences may have arisen for several reasons. First, crop yield was estimated based on heat unit theory, which assumes that the rate of growth is directly proportional to the increase in temperature (Hanasaki et al., 2008). Thailand has a warm climate, causing heat unit theory to slightly overestimate potential

yields. Second, the parameters of the crop module were set according to US standards; these values will differ for Asian countries. Third, no fertilizer stress was applied to the crops in the model. Conversely, the simulated yield of sugarcane was 4–8 t/ha, which was smaller than the observed value of 47.77 t/ha. This difference is presumably because the WAOB-USDA report does not separate yields into irrigated and non-irrigated, while most sugarcane cultivation in Thailand is rainfed. Although the crop growth module of the H08 model predicts crop yields, it was designed primarily for simulating crop

calendars. This further amplified the fluctuations in predicted yield.

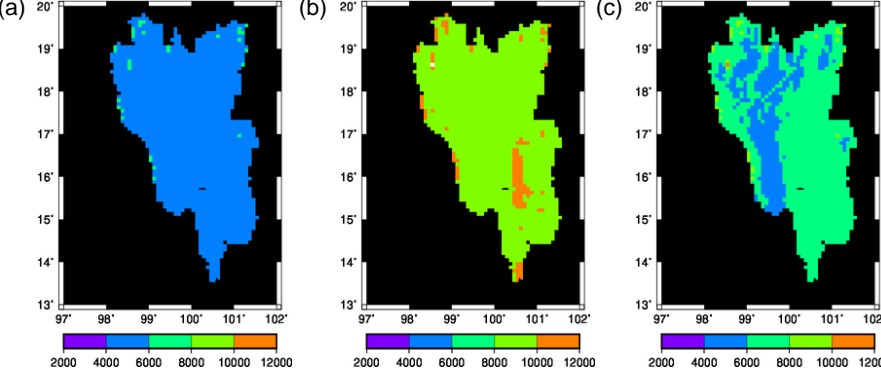

**Figure S5. The simulated annual average yield of the major crops, (a) rice, (b) maize, and (c) sugarcane in kg/ha.**

**S4.3.2 Irrigation water withdrawal**

The coupled H08 model simulates irrigation water withdrawal based on consumptive water use and regional irrigation efficiency. The irrigation efficiency and cropping intensity for the CPRB were set to 50% and 1.5, respectively (Molle et al., 2001; FAO, 2013). Irrigation efficiency of 50% indicates that approximately 50% of the water withdrawn for irrigation becomes delivery losses and return flow. Cropping intensity of 1.5 means that on average 150% of the total irrigated cropland is used for cultivation. Reported irrigation water withdrawal for Thailand is approximately 51.8 km$^3$/year (FAO

2013; Kiguchi et al., 2021), of which nearly 75% (38.9 km$^3$/year) is utilized in the CPRB based on the Water Resources Master Plan produced by the Office of the National Water Resources. Furthermore, four irrigation simulations were

conducted (Table S5) to estimate irrigation water withdrawal in the CPRB in which the virtual inexhaustible and non-renewable water sources were considered in the H08 model to fully meet agricultural water demand and avoid water stress (Hanasaki et al., 2018). Four parameters were used in these simulations (Table S5), representing soil moisture targets for paddy and non-paddy crops. Above this soil moisture threshold, the model assumes no water stress; below this threshold, water stress prevents optimal growth and crop yield (Hanasaki et al., 2008).

Case 1 employed the default parameter values established for the global setup, and the simulated water withdrawal under Case 1 was approximately 74 km$^3$/year. The parameters were slightly adjusted for three additional cases (Case 2, Case 3, and Case 4) to obtain simulated irrigation water withdrawal comparable with the observation (38.9 km$^3$/year). In Case 2, irrigation for non-paddy crops was removed, because paddy is the major irrigated crop in Thailand. However, irrigation water withdrawal remained high, with a value of 51.8 km$^3$/year. The soil moisture target for first paddy crops (rainy-season crops; Fig. S4) was reduced to 0.9 in Case 3, and the simulated water withdrawal (33.7 km$^3$/year) was comparable with the observation. For Case 4, the soil moisture target was further reduced to 0.8 for the first paddy crops, while full irrigation was maintained for the second crops (dry-season crops; Fig. S4). This simulation generated lower irrigation water withdrawal (26.2 km$^3$/year), compared with the observed data.

**Table S5. Validation of the irrigation water withdrawal for the CPRB.**

| | Simulations | Case 1 | Case 2 | Case 3 | Case 4 |
|---|---|---|---|---|---|
| Parameters of crop growth module | Factor for paddy (1$^{st}$ crop) | 1.0 | 1.0 | 0.9 | 0.8 |
| | Factor for paddy (2$^{nd}$ crop) | 1.0 | 1.0 | 1.0 | 1.0 |
| | Factor for non-paddy (1$^{st}$ crop) | 0.75 | 0 | 0 | 0 |
| | Factor for non-paddy (2$^{nd}$ crop) | 0.75 | 0 | 0 | 0 |
| Simulated irrigation water withdrawal (km$^3$/year) | | 74.07 | 51.78 | 33.71 | 26.17 |

The parameters obtained from all four cases were used to simulate irrigated discharge (IRG) by coupling all six modules of the H08 model. In reality, this IRG discharge should correspond to the observed discharge because it includes most of the human interactions such as the reservoir operation and irrigation water abstraction. Therefore, the IRG discharge was compared with the observed discharge at Nakhon Sawan (C.2 station), the calibration point for the CPRB in this study, for final hydrograph reproducibility. The first three cases were later discarded, and Case 4 parameters were employed for further irrigation simulations, because they best reproduced the observed discharge hydrograph at Nakhon Sawan, as shown in Fig. S6. Irrigation water withdrawal in the CPRB for Case 4 was approximately 50% of the reported irrigation water withdrawal

for Thailand; this was acceptable for use in further simulations because specific water withdrawal information was unavailable for the CPRB. The model performed adequately in replicating the observed discharge under Case 4, except for the peak discharge values. The daily and monthly NSE values of the IRG discharge simulation at Nakhon Sawan were 61.93% and 64.58%, respectively.

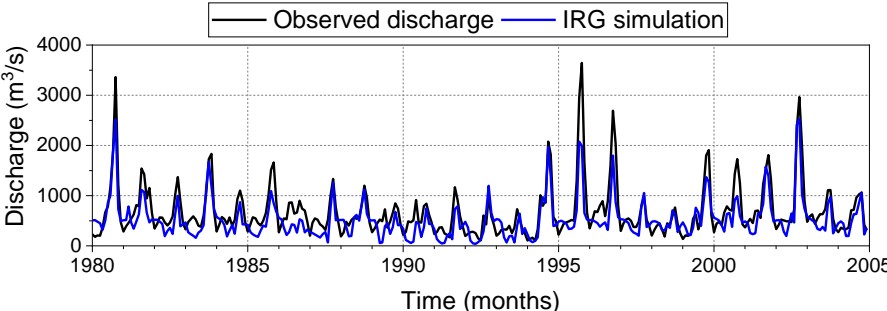

**Figure S6: Monthly hydrograph of IRG simulation compared with observed discharge at Nakhon Sawan.**

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
