# Peer review of "Inclusion of flood diversion canal operation in the H08 hydrological model with a case study from the Chao Phraya River Basin: Model development and validation"

_Hydrology and Earth System Sciences, 2021_

## Referee Comment (RC2)

**Comments on "Inclusion of flood diversion canal operation in the H08 hydrological model with a case study from the Chao Phraya River Basin – Part 1: Model development and validation"**

Integrating anthropogenic factors in GHMs is an important but tough job due to vast varied implementations of water conservation engineering and lack of data. In this study, Gopalan et al., developed a novel diversion module in a GHM H08 and tested its performances as well as other anthropogenic processes in a data-rich river catchment in Thailand. This module enhanced the robustness of GHMs in reproducing hydrological processes under human perturbations. Additional, it provides the fundamental to evaluate the potential of flood control, which is a crucial topic regarding sustainable development under climate change. Although there are several issues needed to clarify before considering for publication, they do not affect the significance of this study and the interests of broad readership.

Major comments

1. Although the introduction underlines the importance of integrating water diversion in GHMs regarding floods control, the structure can be tighter. For instance, Paragraph 2 and 3 (Line 38 — 57) can be combined to introduce the demand of water diversion for flood control and its potential impacts on the water cycle. Paragraph 4 (Line 59 — 72) can be modified by reviewing relevant studies regarding flood control (water quality is trivial) and pointing out the limitation of hydrodynamic models in investigations of earth systems and the lack of water diversion in GHMs.

2. According to the Eq. 2, there may be a jump of $D_{wet}$ (from $Q_{min}$ to $Q-Q_{env}$) when the Q is reaching the $Q_{rivcap}$. Is it what you expected? Additional, a minor issue is that the case $Q<Q_{env}$ is not listed in Eq. 1 or 2.

3. The introduction of the diversion operation (Line 144 — 179) can be clarified more briefly. Instead of explaining each cases listed in Eq. 1 and 2, it is better to describe the logic of operation. The dry season for example, the operation strategy tries to meet the minimum flow diversion ($Q_{min}$) on the premise of guaranteeing the environmental flow ($Q_{env}$).

4.  Two issues are needed to clarify in the calibration.
    a.  The model is calibrated by naturalized and regulated discharges. And irrigated discharge is mentioned as well. How was the discharge naturalized?   Is the regulated discharge equivalent to the irrigated discharge. What is the difference between the two and the observed discharge? Please provide more information.
    b.  The crop module is calibrated the crop module to meet the census irrigated discharge. How is the contribution of diversion canal to irrigation withdrawal in CPRB? If it was significant, the calibrated irrigated discharge (no irrigation due to diversion) must be higher than the observations, correct?
5.  Several questions regarding the physical processes
    a.  Do the canal network and the retention area affect the land surface processes? For example, does the retention area (1-m deep) maintain water during precipitation? Will surface runoff go to canals instead of going to the natural river channel?
    b.  Is the assumption of retention storage cleaning (Line 178 — 179) reasonable? I guess that a part of the water might be loss through deep drainage. Moreover, it is more proper to re-set the retention storage at the end of a hydrological year rather than a normal year?
    c.  Canal water only can be withdrawn by neighbouring grid cells for irrigation. Does it imply that the spatial resolution of simulation will affect the results?
    d.  Seems that the rain-fed cropland is assumed to be used as retention pond. Is it a universal assumption in the future global simulations? If so, will it affect the crop yield?
6.  In Figure 7, the annual average discharge is employed to demonstrate the effect of flood control. I think it is more suitable to present the intra-annual fluctuations of discharge rather than the annual mean, isn't it?
7.  The presentation skill can be further improved. I listed several suggestions below in 'minor comments', but not limited to them.

Minor comments
1.  Line 76: "have been generated by" —> "due to"
2.  Line 110: Do 0.5° and one-day represent the finest resolution that the H08 can reach? If not, please remove it.
3.  Line 144: Please briefly describe the definition of dry/wet season.

4. Line 228 — 233: Please shortly explain what 5, 50, and 90 means in Q_5, Q_50, and Q_90 (% of quantile?).
5. Line 279: "these 5 canals …" It is properer to use word rather than Arabic numeral if the number is less than ten, or less than one hundred if the number represents for the amount of objects. Please check other places as well as the supplement.
6. Line 453 "a great potential" —> "a great potential of flood control".
7. Line 489: "values": What values?
8. Line 490: Remove "simulated".
9. Figure 2: Modify the second sentence to "The green area denote the river basin".
10. Please modify the y-label of Figure 8. It looks like number of days divided by 25 years.
11. Supplement Line 150: Please add the unit of cropping intensity.

---

## Author Comment (AC1)

**Comments on "Inclusion of flood diversion canal operation in the H08 hydrological model with a case study from the Chao Phraya River Basin – Part 1: Model development and validation"**
**Reply to RC1**

This paper describes the successful implementation of flood diversion canal operation in H08 for the Chao Phraya River Basin, which accounts for over half of annual average river discharge diversion of the CPRB. This novel implementation is clever, well described, and I find the paper quite enjoyable to read. I only note a few places where this paper may benefit from improved clarity before publication. Below are a few minor comments/questions for the authors.

Thank you very much for your valuable comments regarding the scientific contribution of our work. We highly appreciate your review comments that provided valuable insights for further modifications of our current version.

The detailed point-by-point replies to all the minor comments are given below. Once again thank you for enlightening us with your valuable comments and suggestions.

**Minor comments**

1. Water diversion during dry season appears to be quite sensitive to the pristine flow simulation used to estimate river and canal carrying capacities. Is it conducted by only including the digitized canal network but excluding direct human influence such as dams, reservoirs and human water use? Are the results then compared with naturalized or raw observed data when computing for NSE (Suppl. S4 L75)?

Thank you very much for your comment. We agree with your point that the water diversion during the dry and wet seasons appears to be quite sensitive to the pristine flow simulation used to estimate river and canal carrying capacities under the generalized scheme. The pristine flow simulation (naturalized simulation in the manuscript; NAT) was conducted by enabling only the land surface hydrology and river routing modules of the H08 model, which does not account for the direct human influences such as dams, reservoirs, and human water use. Further, this "NAT" simulation was compared with the 'naturalized observed discharge' at Nakhon Sawan (C.2 station). The 'naturalized observed discharge' at Nakhon Sawan was reconstructed by

following Mateo et al. (2014). This was performed by adding the water stored in the two major dam reservoirs (Bhumibol and Sirikit) with the 'observed discharge' at Nakhon Sawan. The 'naturalized observed discharge' was adequately reproduced at Nakhon Sawan, with daily and monthly Nash-Sutcliffe efficiency (NSE) values of 75.18% and 86.07%, respectively. Estimating the 'naturalized observed discharge' downstream of Nakhon Sawan station must be extremely difficult because of the presence of many unmonitored canals. The transformation of 'observed discharge' into the 'naturalized observed discharge' and the associated uncertainties are described in detail by Champathong et al. (2020). These explanations were added to section S4.1 of the supplementary material to avoid confusion. Line 76

Mateo, C. M., Hanasaki, N., Komori, D., Tanaka, K., Kiguchi, M., Champathong, A., Sukhapunnaphan, T., Yamazaki, D. and Oki, T.: Assessing the impacts of reservoir operation to floodplain inundation by combining hydrological, reservoir management, and hydrodynamic models, Water Resour. Res., 50(9), 7245–7266, https://doi.org/10.1002/2013WR014845, 2014.

Champathong, A., Hanasaki, H., Kiguchi, M. and Oki, T.: Reconstructing the pristine flow of highly developed rivers − a case study on the Chao Phraya River, Hydrol. Res. Lett., 14(2), 89-96. https://doi.org/10.3178/hrl.14.89, 2020.

I am wondering why the river canal capacity decreases along the natural river channel between some locations (i.e., C.13 to C.3), even when there are no canals between them (Figure 4). Also, it seems that the capacity values shown in Figure 4 are a mix of simulated Q5 (main river) and observed (canal, Table S1). Is this correct? It would be quite informative if the simulated river/canal carrying capacities are also listed in Table S1.

We would like to clarify that the river and canal carrying capacities shown in Fig. 4 are the observed values in the CPRB. These carrying capacities of the river channel and canals at various locations are solely determined by their cross-sections. Near the Chao Phraya dam (C.13 station), the river channel can hold a maximum discharge of 2840 m$^3$/s, whereas at Sing Buri (C.3 station) the channel gets narrower and can hold a maximum discharge of 2340 m$^3$/s. Since these are observed values, the presence or absence of canals does not play any role in the river carrying capacities at these locations. Sincere apologies for the confusion made by us. To avoid further confusion, we have modified the caption of Fig. 4 as well as section 3.2.1 of the manuscript that the values shown in Fig. 4 are observed values. Line 283; Line 306; Line 809

In addition, the simulated river and canal carrying capacities under the regionalized and generalized schemes are also included in Table S1 to provide more clarity. Line 371 (manuscript); Line 9 (supplementary material)

2. While the generalized canal scheme has potential for global applications, a major obstacle is the estimation of retention areas. In this study paddy fields were used as retention area with fixed depth, and this would not be applicable globally. I am curious how the authors would apply this scheme globally, especially when the bathymetry of lakes/ponds are not known and cannot use the 1 m depth assumption.

Thank you very much for your comment. The most important land use for potential retention areas is the low-lying areas along rivers (floodplains) and canals. Historically, such lowland is used for paddy cultivation in warm Asian countries. Being paddy is not the required condition for retention areas. In addition, although the lakes/ponds could be partially filled with water during the wet season, they can also be used as retention areas based on available free space. Under such circumstances, the bathymetry of lakes/ponds may be useful but not essential for estimating potential areas for the retention pond. Indeed, some of them are permanently inundated (i.e., maintained by groundwater flow, etc.) and hence cannot be used as effective retention storage.

For modelling, the geographic locations of possible retention areas (e.g., low-lying areas, lakes, ponds, wetlands, etc.) along with their depth and areal extents available for storage of floodwater specific to each area should be estimated. This information can be extracted from remotely sensed data such as general DEMs (e.g., MERIT DEM), satellite imageries (MODIS/LANDSAT), radar altimetry, as well as from literature although it is strenuous. There are several databases (G-REALM, HYDROWEB, RLH, DAHITI, etc.) from which we can extract the information regarding lakes/ponds. The global application of this scheme that includes the estimation of retention areas is one of the limitations this study currently poses, and we will pursue further research into this area. This explanation was added to the discussion (section 5 of the manuscript) to have more clarity. Line 578

3. Although the authors already did a fantastic job describing the model, I would still like to ask a few questions to make sure I understand the details correctly: In P11L346, "10% of diverted water is supplied to each of the nearby grid cells that was further utilized for irrigation". Do you mean for each of the 5' grid that the canal passes, 10% of total diverted

water is supplied to that grid? So as the diverted water flows along canal to each grid, it first loses 10% of water for water supply, then fully fill that grid's retention capacity before moving to next grid, where this process is repeated until either the water is fully contained in the retention area, or flows out of the basin. Is this correct (I am especially uncertain about the retention filling: P14L441 says "this runoff constitutes a portion of retention pond storage")? If water is only supplied to grids the canal flows to, then the schematic diagram of Figure 2 should perhaps be slightly modified and remove the second arrow of B on the lower left. Also, how is the water balance closed if irrigation demand is less than water supplied to the local grid? And would this "supply to nearby grid" percentage change if the simulation is performed on finer/coarser resolution?

Thank you very much for your comment. The operation of the canal system introduced in this study depends upon the dry and wet seasons. During the dry season, a minimum amount of water is diverted into the canals. Once diverted into the canals, 10% of the diverted water is supplied to each of the $5' \times 5'$ grid cells through which the canal passes as well as to the immediate lateral neighbouring grid cells of the canal. This water is used to meet the irrigation demand. If the demand is less than the water supplied to the local grid, then the surplus water after meeting the demand is further added to the discharge of the corresponding grid cell. This river discharge finally returns to the river channel as shown in Fig. S1a and b and thereby closes the water balance. The remaining diverted water after supply will move to the subsequent downstream grid cells. This process is repeated until the diverted flow is fully depleted or reaches its destination. This supply component is enabled only during the dry season to augment water supply needs. In this study, for simplicity, 10% of diverted water is supplied to each of the nearby grid cells because our primary concern was flood control. Therefore, of course, we should change this fraction of 'supply to near grids' if we are performing the simulation on a finer/coarser resolution. One alternative way to overcome this issue is that we can finalize the 'supply to near grids' based on the water demand in each of the grid cells through which the canal passes as well the in the neighbouring grid cells. In such instances, it can be confirmed that the supplied water will be completely utilized. These explanations were added to the manuscript to avoid confusion regarding the 'supply to near grids' component (section 2.2 and section 5). In addition, Fig. 2 has been slightly modified to clearly portray the water supply to the grid cell through which the canal passes as well as to the immediate lateral neighbouring grid cells. Line 156; Line 190; Line 570; Line 802

During the wet season, either canal carrying capacity, or a minimum amount of flow is diverted to the canals. Once diverted, a portion of the diverted flow drains into the retention areas and then fills to the grid's retention pond capacity before moving to the next grid. This process is repeated along its flow route until flow either diminishes to zero or reaches its destination (either within the basin or out of the basin). The storage of diverted water in retention areas is allowed only during the wet season to supplement flood control. In addition to the diverted water storage during the wet season, the retention areas are modelled in such a way that they receive runoff generated from precipitation in each grid based on their areal fraction during both dry and wet seasons. This runoff constitutes a part of retention pond storage and only the remaining storage capacity is available for the storage of diverted floodwater during the wet season. These explanations were added to the manuscript (section 2.2) to avoid confusion regarding the 'retention storage' component. Line 201

4. What are the similarities between the explicit aqueduct water transfer module and this canal operation module?

Thank you very much for your comment. The earlier aqueduct module of the H08 model was to provide water supply to the grid cells that are farther from the river channel to meet their water demand (agricultural, industrial, and domestic) through structures of canals, pipes, and others. If there is a water demand to meet, the scheme assumes that the water could be transferred until the river flow at the aqueduct origin falls below the environmental flow because the information regarding the aqueduct carrying capacity was not available for most cases (Hanasaki et al., 2018). This aqueduct water transfer scheme transfers water only when the water demand is positive. It does nothing for excess water availability (i.e., floodwater). To overcome this limitation, we introduced the new canal operation scheme. This scheme operates to provide a minimum water supply during the dry season irrespective of the water demand and divert floodwater (subject to a maximum of the canal carrying capacity) during the wet season to reduce flood risk. In both cases, environmental flow is maintained in the river channel. To have a clear differentiation between the aqueduct water transfer scheme and the newly introduced canal operation scheme, more explanations were added regarding the operation of the aqueduct water transfer scheme of the H08 model in the manuscript (section 2.2). Line 121

Hanasaki, N., Yoshikawa, S., Pokhrel, Y. and Kanae, S.: A global hydrological simulation to specify the sources of water used by humans, Hydrol. Earth Syst. Sci., 22(1), 789–817, https://doi.org/10.5194/hess-22-789-2018, 2018.

How do you determine if it is canal or aqueduct based on Google Earth images?

For global applications, recently, Shumilova et al. (2018) prepared a global inventory of 110 water transfer megaprojects (exiting, planned, and proposed) from which the canal origin, destination, route, purpose, type of canal, and carrying capacity can be retrieved. During global applications, such kinds of global inventories can be utilized to get the canal information. Since the H08 model does not consider the hydraulic characteristics of the water conveying structure, it can be assumed as an open channel, pipe, or any other structure if the data on aqueduct type is not available. In a similar fashion, we introduced a generalized scheme that operates with $Q_{50}$ as the canal carrying capacity (with the assumption that the median flow should be diverted under flood conditions) under the limited data availability scenario.

Shumilova, O., Tockner, K., Thieme, M., Koska, A. and Zarfl, C.: Global Water Transfer Megaprojects: A Potential Solution for the Water-Food-Energy Nexus?, Front. Environ. Sci., 6(DEC), 150, https://doi.org/10.3389/fenvs.2018.00150, 2018.

5. Figure S4: Are you using multi-year averaged crop calendar?

Thank you very much for your comment. The authors would like to make clear that the crop calendar is not multi-year averaged. Instead of estimating crop calendar for every single year and generating a multi-year averaged crop calendar, we multi-averaged the variables (air temperature, shortwave downward radiation, evapotranspiration, and potential evapotranspiration) that were used to compute the crop calendar from 1980 to 2004. Then by utilizing these multi-year averaged variables, we estimated a single crop calendar for each of the crops in CPRB. Later, we compared this simulated crop calendar with the observed crop calendar of major crops in Thailand (Fig. S4) and the planting and harvesting dates were fairly captured. In order to avoid confusion, we added this explanation to the Supplementary material (S4.3.1). Line 120

---

## Author Comment (AC2)

**Comments on "Inclusion of flood diversion canal operation in the H08 hydrological model with a case study from the Chao Phraya River Basin – Part 1: Model development and validation"**
**Reply to RC2**

Integrating anthropogenic factors in GHMs is an important but tough job due to vast varied implementations of water conservation engineering and lack of data. In this study, Gopalan et al., developed a novel diversion module in a GHM H08 and tested its performances as well as other anthropogenic processes in a data-rich river catchment in Thailand. This module enhanced the robustness of GHMs in reproducing hydrological processes under human perturbations. Additional, it provides the fundamental to evaluate the potential of flood control, which is a crucial topic regarding sustainable development under climate change. Although there are several issues needed to clarify before considering for publication, they do not affect the significance of this study and the interests of broad readership.

Thank you very much for your valuable comments regarding the scientific contribution of our work. We highly appreciate your review comments that provided valuable insights for further modifications of our current version.

The detailed point-by-point replies to all the major as well as the minor comments are given below. Once again thank you for enlightening us with your valuable comments and suggestions.

**Major comments**

1. Although the introduction underlines the importance of integrating water diversion in GHMs regarding floods control, the structure can be tighter. For instance, Paragraph 2 and 3 (Line 38 — 57) can be combined to introduce the demand of water diversion for flood control and its potential impacts on the water cycle. Paragraph 4 (Line 59 — 72) can be modified by reviewing relevant studies regarding flood control (water quality is trivial) and pointing out the limitation of hydrodynamic models in investigations of earth systems and the lack of water diversion in GHMs.

Thank you very much for your comment. According to the comment, the authors have tightened the structure of the Introduction section further to remove redundancies and improve the understanding. Once again thank you very much for the nice idea. Line 27

2. According to the Eq. 2, there may be a jump of D_wet (from Q_min to Q- Q_env) when the Q is reaching the Q_rivcap. Is it what you expected?

Thank you very much for your valuable comment. It is true that there will be a jump in the daily $D_{wet}$ values from $Q_{cancap}$ to $Q_{min}$ when the river discharge falls below the $Q_{rivcap}$. There will be another jump in $D_{wet}$ values from $Q_{min}$ to Q-$Q_{env}$ when river discharge is approaching the environmental flow. However, this pattern of sudden changes in canal flow was also seen in the observed canal diversion flow values (please see the below figure for reference). We tried to replicate this observed pattern using the proposed canal operations.

[Figure]

Figure 1. Observed canal discharge for different canal systems during the wet season of 2016.

Additional, a minor issue is that the case Q<Q_env is not listed in Eq. 1 or 2.

Thank you very much for notifying the error that we made in Eq. 1 and 2. According to the comment, we have added the case of canal flow diversion when Q< $Q_{env}$ in Eq. 1 and 2. Line 147; Line 167

3. The introduction of the diversion operation (Line 144 — 179) can be clarified more briefly. Instead of explaining each cases listed in Eq. 1 and 2, it is better to describe the logic of operation. The dry season for example, the operation strategy tries to meet the minimum flow diversion (Q_min) on the premise of guaranteeing the environmental flow (Q_env).

Thank you very much for your valuable comment. According to the comment, the logic of operation during the dry/wet season has been explained briefly rather than explaining each case listed in Eq. 1 and 2. Thank you for providing this nice idea. The modified sections are as follows:

*"The dry season is characterized by low rainfall that causes consequent water shortage for an extended period. To alleviate the water scarcity issues, water supply should be provided from the river channel to the neighbouring areas through diversion canals by preserving the environmental flow in the river channel. Based on this aspect, the operation of diversion canal systems during the dry season leads to the emergence of three cases concerning the environmental flow ($Q_{env}$) required to be maintained in the river channel, which are expressed as follows:*

$$D_{dry} = \begin{cases} Q_{min} & ; if\ Q > Q_{env}\ and\ (Q - Q_{min}) > Q_{env} \\ Q - Q_{env} & ; if\ Q > Q_{env}\ and\ (Q - Q_{min}) < Q_{env} \\ 0 & ; if\ Q < Q_{env} \end{cases} \qquad (1)$$

*where $D_{dry}$ is the daily water diversion during the dry season; $Q_{min}$ is minimum flow diversion; $Q_{env}$ is the environmental flow requirement; and $Q$ is the daily river discharge at the origin of diversion. The first two cases in Eq. 1 represent two low flow scenarios (Supplementary Fig. S1a and b) and can be explained as follows: (i) the operation strategy tries to meet minimum flow diversion ($Q_{min}$) on the premise of guaranteeing the environmental flow ($Q_{env}$) in the river channel even after water diversion due to enough water availability, and (ii) the diversion criterion attempts to divert water ($Q - Q_{env}$) that is smaller in quantity when compared to the $Q_{min}$ to ensure the required the environmental flow requirement in the river channel due to the relatively low water availability. Using these diversion criteria, environmental flow is maintained in both cases. No flow is diverted to the canal if river discharge is lower than environmental flow, depicted as the third case in Eq. 1."* Line 142

*"The wet season is the period during which most of the annual rainfall is received. This high rainfall will eventually cause flooding in the neighbouring areas whenever river discharge exceeds the river channel carrying capacity. The diversion canals can divert this floodwater from the river channel and restore the river water level below the carrying capacity. From this perspective, five relevant cases can be identified for the operation of diversion canals during the wet season, as shown below:*

$$D_{wet} \begin{cases} Q_{cancap} & ; if\ Q > Q_{rivcap}\ and\ (Q - Q_{cancap}) > Q_{env} \\ Q - Q_{env} & ; if\ Q > Q_{rivcap}\ and\ (Q - Q_{cancap}) < Q_{env} \\ Q_{min} & ; if\ Q < Q_{rivcap}\ and\ (Q - Q_{min}) > Q_{env} \\ Q - Q_{env} & ; if\ Q < Q_{rivcap}\ and\ (Q - Q_{min}) < Q_{env} \\ 0 & ; if\ Q < Q_{env} \end{cases} \qquad (2)$$

*where $D_{wet}$ is the daily water diversion during the wet season; $Q_{cancap}$ is the maximum canal carrying capacity; and $Q_{rivcap}$ is the river channel carrying capacity. The first four cases in Eq. 2 are based on whether river discharge ($Q$) is greater or smaller than the river channel carrying capacity ($Q_{rivcap}$). The first two out of the four cases correspond to the flood flow scenario, where $Q > Q_{rivcap}$ (Supplementary Fig. S1c and d). In the flood flow scenario, the operation strategy tries to divert the maximum possible amount of floodwater that is equivalent to the $Q_{cancap}$ (Case I of Eq. 2) to keep the river flow below the $Q_{rivcap}$. However, at some point, the river flow in the main channel falls below the $Q_{env}$ after the diversion of $Q_{cancap}$. In such instances, the remaining river discharge after meeting the environmental flow requirement ($Q - Q_{env}$) is diverted into the canal instead of diverting the $Q_{cancap}$ (Case II of Eq. 2). The last two out of the four cases represent non-flood flow scenario, where $Q < Q_{rivcap}$ (Supplementary Fig. S1e and f). Although the river discharge lies below the $Q_{rivcap}$ in the non-flood scenario, a minimum flow ($Q_{min}$) is diverted from the main channel to reduce flooding at downstream locations (Case III of Eq. 2). If the diversion of $Q_{min}$ to the canal reduces the river discharge below the $Q_{env}$, then a decreased quantity of water ($Q - Q_{env}$) is diverted from the main channel rather than the $Q_{min}$ to maintain the environmental flow in the river (Case IV of Eq. 2). Canal diversion remains zero during periods of environmental flow (Case V of Eq. 2)."* Line 162

4. Two issues are needed to clarify in the calibration.

   a) The model is calibrated by naturalized and regulated discharges. And irrigated discharge is mentioned as well. How was the discharge naturalized?

   Thank you very much for your valuable comment. First of all, the authors would like to apologize for the confusion made by us. To avoid further confusion, we have used acronyms in the revised manuscript to represent all the five simulations carried out in this study. The conducted simulations are naturalized (NAT; no human influences), regulated (DAM; dam operation is enabled), irrigated (IRG; dam + water withdrawal for irrigation is

enabled), regionalized (REG; dam + water withdrawal for irrigation + regionalized canal scheme is enabled), and generalized (GEN; dam + water withdrawal for irrigation + generalized canal scheme is enabled) as shown in Table 3. The regulated simulation was renamed as "dam simulation" to provide more clarity in the revised manuscript. Line 392; Line 834

Regarding the model calibration, the H08 model was calibrated for pristine flow at Nakhon Sawan and further validated at various stations in the CPRB. To do so, initially, the "NAT" discharge was simulated by enabling the land surface hydrology and river routing modules of the H08 model, which do not include the effect of water infrastructures and thereby simulate the pristine flow. Further, this "NAT" simulation was compared with the 'naturalized observed discharge' at Nakhon Sawan (C.2 station). The 'naturalized observed discharge' at Nakhon Sawan was reconstructed by following Mateo et al. (2014). This was performed by adding the water stored in the two major dam reservoirs (Bhumibol and Sirikit) with the 'observed discharge' at Nakhon Sawan. These explanations were added to the manuscript (section 4) and supplementary material (section S4.1) to have more clarity. Line 409 (manuscript); Line 76; Line 102 (supplementary material)

Mateo, C. M., Hanasaki, N., Komori, D., Tanaka, K., Kiguchi, M., Champathong, A., Sukhapunnaphan, T., Yamazaki, D. and Oki, T.: Assessing the impacts of reservoir operation to floodplain inundation by combining hydrological, reservoir management, and hydrodynamic models, Water Resour. Res., 50(9), 7245–7266, https://doi.org/10.1002/2013WR014845, 2014.

> Is the regulated discharge equivalent to the irrigated discharge. What is the difference between the two and the observed discharge? Please provide more information.

In addition to the pristine flow prediction, the ability of the calibrated parameters in reproducing the observed discharge at the Nakhon Sawan was assessed. For this purpose, initially, the "DAM" discharge was simulated by enabling the reservoir operation module of the H08 model in addition to the land surface hydrology and river routing modules. Then, the "DAM" discharge and observed discharge were compared at Nakhon Sawan under the assumption that the "DAM" simulation could act as a proxy for the observed discharge although precisely not the case because the "DAM" discharge simulation does not include water abstraction for irrigation. Still, this comparison was made to evaluate the performance of the included reservoir operations in the model. Lastly, all six modules of

the H08 model were enabled to simulate the "IRG" discharge. In fact, this "IRG" simulation should correspond to the observed discharge because it includes most of the human interactions such as the reservoir operation and irrigation water abstraction. Therefore, the "IRG" simulation was compared with the observed discharge for final hydrograph reproducibility. Both the "DAM" and "IRG" discharge simulations were compared with the same observed discharge to examine the performance of different human interactions (reservoir operation and irrigation water abstraction) with the water cycle in terms of hydrograph reproducibility. The authors sincerely apologize for the confusion made by us and this comment really helped us to improve the clarity and understanding of the explanations regarding the naturalized (NAT), dam (DAM), and irrigated (IRG) discharge simulations in the manuscript. According to the comment, these explanations were added to the supplementary material (sections S4.2 and S4.3.2) to provide a better understanding. Line 102; Line 192

b) The crop module is calibrated the crop module to meet the census irrigated discharge. How is the contribution of diversion canal to irrigation withdrawal in CPRB?

Thank you very much for your comment. The actual contribution of diversion canal to irrigation withdrawal in CPRB is unknown because water taken from rivers and canals is seldom reported separately. Reported irrigation water withdrawal for Thailand is approximately 51.8 $km^3$/year (FAO 2013; Kiguchi et al., 2021), of which nearly 75% (38.9 $km^3$/year) is utilized in the CPRB based on the Water Resources Master Plan produced by the Office of the National Water Resources. This includes the contribution from both river and canal systems, whose individual share is unknown.

FAO (Food and Agriculture Organization): AQUASTAT Core Database. Food and Agriculture Organization of the United Nations. http://www.fao.org/aquastat/en/, Database accessed on 2021/04/19, 2013.

Kiguchi, M., Takata, K., Hanasaki, N., Archevarahuprok, B., Champathong, A., Ikoma, E., Jaikaeo, C., Kaewrueng, S., Kanae, S., Kazama, S., Kuraji, K., Matsumoto, K., Nakamura, S., Nguyen-Le, D., Noda, K., Piamsa-Nga, N., Raksapatcharawong, M., Rangsiwanichpong, P., Ritphring, S., Shirakawa, H., Somphong, C., Srisutham, M., Suanburi, D., Suanpaga, W., Tebakari, T., Trisurat, Y., Udo, K., Wongsa, S., Yamada, T., Yoshida, K., Kiatiwat, T. and Oki, T.: A review of climate-change impact and adaptation

studies for the water sector in Thailand, Environ. Res. Lett., 16(2), 023004, https://doi.org/10.1088/1748-9326/abce80, 2021.

If it was significant, the calibrated irrigated discharge (no irrigation due to diversion) must be higher than the observations, correct?

In the H08 model, the crop growth module mainly simulates the crop calendar and crop yields. The irrigation water demand is estimated by the coupled module of the H08 model only when there is a soil moisture deficit. The soil moisture should maintain at 75% of field capacity for non-paddy crops and 100% of field capacity for paddy crops in irrigated fields. There will be a soil moisture deficit below this threshold and water is withdrawn from the river to meet this deficit. The simulated irrigation water withdrawal by the coupled model for the CPRB was nearly 26 $km^3$/year (nearly 50% of water withdrawal for Thailand) in comparison with the observed withdrawal of 39 $km^3$/year (Supplementary material S4.3.2). While simulating the irrigated (IRG) discharge, the canal operations were not enabled, and the agricultural water demand has met mainly by the water withdrawal from the river.

Once the canal operations were enabled, that simulation was referred to as the regionalized discharge (REG) in this study because it includes region-specific canal operation. During this canal operation, a huge amount of water is diverted from the river channel (≈13 $km^3$/year; Fig. 6) in which only 1.6 $km^3$/year is utilized for irrigation due to the absence of irrigation supply data from canals. This amount is not significant compared with the simulated water withdrawal of 26 $km^3$/year. Most of the remaining diverted water is returned to the river channel (9.6 $km^3$/year) and the remainder is taken out of the basin or stored in retention areas (2 $km^3$/year). This returned water to the river channel is further available for irrigation water abstraction because the order of water withdrawal to meet the agricultural water demand assumes that the priority should be given to the canal water supply. If the canal water supply is not meeting the water demand fully, the water is withdrawn from the river system to meet the demand gap. Due to this water withdrawal from rivers, the calibrated irrigated discharge will not be higher than the observations.

5. Several questions regarding the physical processes

   a) Do the canal network and the retention area affect the land surface processes? For example, does the retention area (1-m deep) maintain water during precipitation? Will surface runoff go to canals instead of going to the natural river channel?

Thank you very much for your comment. The retention areas will affect the land surface processes, but not the canal network. The retention areas are modeled in such a way that they receive runoff generated from precipitation in each grid based on their areal fraction during both dry and wet seasons. This runoff constitutes a part of retention pond storage and only the remaining storage capacity is available for the storage of diverted floodwater during the wet season. In the case of canal networks, it was assumed that the surface runoff will not go to the canals, and they only receive diverted water from the river network. These explanations on how the retention areas can affect the land surface processes were added to the manuscript to avoid confusion. Line 204

b) Is the assumption of retention storage cleaning (Line 178 — 179) reasonable? I guess that a part of the water might be loss through deep drainage.

Thank you very much for your comment. Of course, the evaporation of retention storage at the end of every normal year is an assumption for simplicity at this stage. In CPRB, the rainfed paddy croplands (retention areas in this study) situate in the low-lying area which is lower than the water level of the rivers and canals (Jamrussri et al., 2018; JICA, 2013). Therefore, pumping stations were constructed to drain the floodwater stored in the paddy fields to the canals immediately after the floods and further operated to drain water from canals to main rivers (JICA, 2013). This will prepare the paddy fields for cultivation and further storage of water if floods occur withinn the same year. However, this pumping process was not modelled in this study due to the challenges involved. We are further developing the operation scheme to include the pumping process for the quick withdrawal of the water stored in the retention areas to represent the real processes. In addition, as the reviewer stated, a part of the retention storage might be lost through percolation. However, due to limitations in the availability of such data, we assumed that the percolation loss will be negligible in the highly saturated paddy fields at the time of floods. A similar assumption was also made for the canal operations that the water will transfer without any loss and delay. To make these physical processes clearer to the readers, these explanations were added to the discussion (section 5) as the uncertainties and limitations involved in the study. Thank you very much for reminding us about this. Line 555

Jamrussri, S., Toda, Y. and Tsubaki, R.: Integrated flood countermeasures in the upper and middle Chao Phraya River Basin, J. Appl. Water Eng. Res., 7(2), 143–155, https://doi.org/10.1080/23249676.2018.1497559, 2018.

JICA (Japan International Cooperation Agency): Project for the comprehensive flood management plan for the Chao Phraya River basin, Final report, Volume 1: Summary Report, CTI Engineering International Co., Ltd., Oriental Consultants Co., Ltd., Nippon Koei Co., Ltd., CTI Engineering Co., Ltd., available at: https://openjicareport.jica.go.jp/pdf/12127205.pdf (last access: 20 April 2021), September 2013.

Moreover, it is more proper to re-set the retention storage at the end of a hydrological year rather than a normal year?

If we reset the retention storage at the end of a hydrological year (March), the water will be kept in these retention areas until then and the paddy fields become not available for the cultivation of the second crop during the dry season (supplementary Fig. S4). Also, we believe that the retention pond or shallow and ill-managed storage cannot keep water until the end of the hydrological year (March). Therefore, due to the above-mentioned reasons as well as the immediate withdrawal of retention storage using pumps, we reset the retention storage at the end of a normal year rather than a hydrological year.

c) Canal water only can be withdrawn by neighbouring grid cells for irrigation. Does it imply that the spatial resolution of simulation will affect the results?

Thank you very much for your comment. Yes, the spatial resolution of the simulation will affect the results. In this study, for simplicity, 10% of diverted water is supplied to each of the nearby grid cells because our primary concern was flood control. Therefore, of course, we should change this fraction of 'supply to near grids' if we are performing the simulation on a finer/coarser resolution. One alternative way to overcome this issue is that we can finalize the 'supply to near grids' based on the water demand in each of the grid cells through which the canal passes as well the in the neighbouring grid cells. In such instances, it can be confirmed that the supplied water will be completely utilized. To make these physical processes clearer to the readers, these explanations were added to the discussion section as the uncertainties and limitations involved in the study. Line 570

d) Seems that the rain-fed cropland is assumed to be used as retention pond. Is it a universal assumption in the future global simulations? If so, will it affect the crop yield?

Thank you very much for your comment. In this study, we assumed that the rainfed cropland can be used as retention areas because, in Thailand, the rainfed cropland comprises low-lying paddy fields that are natural floodplains and cannot be cultivated during the rainy season because of flooding (Jamrussri et al., 2018). This assumption will affect the crop yield because the cropland is not available for cultivation until the beginning of the subsequent year due to the storage of floodwater. This is one of the limitations this study currently poses and that will be addressed in our future studies by considering the actual pumping process (please refer to the response of major comment 5b) in the canal operation scheme for the CPRB. However, this is not a universal assumption for future global applications. Being the rainfed cropland is not the required condition for retention areas. The most important land use for potential retention areas is the low-lying areas along rivers (floodplains) and canals. In addition, although the lakes/ponds could be partially filled with water during the wet season, they can also be used as retention areas based on available free space. For modelling, the geographic locations of possible retention areas (e.g., low-lying areas, lakes, ponds, wetlands, etc.) along with their depth and areal extents available for storage of floodwater specific to each area should be estimated. This information can be extracted from remotely sensed data such as general DEMs (e.g., MERIT DEM), satellite imageries (MODIS/LANDSAT), radar altimetry, as well as from literature although it is strenuous. In this study, the authors tried to propose a way to set parameters from simulations for the future global application (generalized canal scheme) rather than to claim that the model applies universally. These explanations were added to the discussion section to have more clarity. Line 578

Jamrussri, S., Toda, Y. and Tsubaki, R.: Integrated flood countermeasures in the upper and middle Chao Phraya River Basin, J. Appl. Water Eng. Res., 7(2), 143–155, https://doi.org/10.1080/23249676.2018.1497559, 2018.

6. In Figure 7, the annual average discharge is employed to demonstrate the effect of flood control. I think it is more suitable to present the intra-annual fluctuations of discharge rather than the annual mean, isn't it?

Thank you very much for your valuable comment. We highly agree with your comment that it is better to present the intra-annual fluctuations of discharge to have a clear idea about the effect of canal operations on flood control. Therefore, according to the comment, we have depicted

the annual and seasonal variations of river discharge under various simulations in Fig. 7. Explanations were also added to support the modified Fig. 7 as follows:

*"Initially, the impact of canal systems and retention areas on reducing the annual average, wet season, and dry season discharges of the CPRB was analyzed. Fig. 7 shows the annual, wet season, and dry season discharges in the CPRB for various simulations averaged from 1980 to 2004. The maximum annual average discharge under the NAT simulation was approximately 850 $m^3$/s in the basin (Fig. 7a1), which may lead to devastating impacts in the lower basin, including Bangkok City. The effect of reservoir operation on annual average discharge was negligible (Fig. 7b1). A marked reduction in discharge, with values ranging between 500 $m^3$/s and 583 $m^3$/s, occurred after enabling irrigation water abstraction (Fig. 7c1). The impact of water diversion on annual average discharge shows that diversion has a great potential for flood control in the lower CPRB (Fig. 7d1 and e1). In the REG simulation, the annual average discharge of the CPRB was approximately 523 $m^3$/s, a reduction of 10% from the IRG simulation. In contrast, the GEN simulation portrayed a reduction of 28% in basin annual average discharge, compared with the IRG simulation.*

*The discharge reduction under various simulations during the wet season was very similar to the annual average flow reduction pattern except for the DAM simulation. Under the DAM simulation, remarkable discharge reduction was observed in the Ping, Nan, and Chao Phraya rivers (Fig. 7b2) due to the operation of upstream dam reservoirs of Bhumibol and Sirikit with an outlet discharge deduction of nearly 15%. The subsequent discharge simulations of IRG, REG, and GEN ones illustrated nearly 34%, 10%, and 26% reduction in the outlet discharge of the CPRB (Fig. 7c2-e2), which further revealed the dominance of wet season irrigation water abstraction and canal operations in the annual pattern. During the dry season, the outlet discharge was nearly 202 $m^3$/s under the NAT simulation (Fig. 7a3), which was enhanced to 551 $m^3$/s after enabling the reservoir operation (Fig. 7b3). Likewise the wet season, further reductions in discharge was noted in the river channel due to the irrigation water abstraction and canal operations (Fig. 7c3-e3). Overall, the canal operations significantly reduced the main channel discharge both annually and intra-annually."* Line 472; Line 820

7. The presentation skill can be further improved. I listed several suggestions below in 'minor comments', but not limited to them.

Thank you very much for your comment. According to the comment, the authors have thoroughly read the manuscript and it has been modified to improve the presentation wherever

necessary. In addition, the authors have addressed all the minor comments raised by the reviewer whose detailed responses can be seen in the below section. The authors are thankful to the reviewer for your thoughtful comments.

**Minor comments**

1.  Line 76: "have been generated by" —> "due to"

Response to comment No. 1: Thank you very much for your comment. According to the comment, "have been generated by" has been changed to "are due to". Line 79

2.  Line 110: Do 0.5° and one-day represent the finest resolution that the H08 can reach? If not, please remove it.

Response to comment No. 2: Thank you very much for your comment. In the H08 model, 0.5° × 0.5° and one day are the finest resolutions for standard global applications. However, the model was designed for application to any spatial resolution as we carried out in this study (a spatial resolution of 5 arcmins). Therefore, in order to avoid confusion, the sentence was rewritten as *"Each of the modules can run separately with standard spatial and temporal resolutions of 0.5° × 0.5° and one day, respectively, on a global scale".* Line 104

3.  Line 144: Please briefly describe the definition of dry/wet season.

Response to comment No. 3: Thank you very much for your comment. According to the comment, we have added a brief definition of dry/wet season as follows:

*"The dry season is characterized by low rainfall that causes consequent water shortage for an extended period."* Line 142

*"The wet season is the period during which most of the annual rainfall is received."* Line 162

4.  Line 228 — 233: Please shortly explain what 5, 50, and 90 means in Q_5, Q_50, and Q_90 (% of quantile?).

Response to comment No. 4: Thank you very much for your comment. According to the comment, we have added a brief description of the $Q_5$, $Q_{50}$, and $Q_{90}$ indices as follows:

*"$Q_5$ value (the 95-percentile flow, which was equalled or exceeded for 5% of the flow record – a high flow representation)"* Line 247

*"Q$_{50}$ value (the 50-percentile flow, which was equalled or exceeded for 50% of the flow record – a medium flow representation)"* Line 250

*"Q$_{90}$ value (the 10-percentile flow, which was equalled or exceeded for 90% of the flow record – a low flow representation)"* Line 253

5. Line 279: "these 5 canals ..." It is properer to use word rather than Arabic numeral if the number is less than ten, or less than one hundred if the number represents for the amount of objects. Please check other places as well as the supplement.

Response to comment No. 5: Thank you very much for your comment. According to the comment, all the Arabic numeral representation of the numbers in the whole manuscript and supplementary file has changed to word format whenever the number is less than ten as well as less than one hundred if the number represents the amount of objects. Thank you for notifying this. The modifications are underlined throughout the manuscript and supplementary file. Line 305 and elsewhere in manuscript and supplementary material

6. Line 453 "a great potential" —> "a great potential of flood control".

Response to comment No. 6: Thank you very much for your comment. According to the comment, "a great potential" has been changed to "a great potential for flood control" for clarity. Line 478

7. Line 489: "values": What values?

Response to comment No. 7: Thank you very much for your comment. According to the comment, "values" has been changed to "the diverted river flow values" for clarity. Line 524

8. Line 490: Remove "simulated".

Response to comment No. 8: Thank you very much for your comment. According to the comment, "simulated" has been removed. Line 524

9. Figure 2: Modify the second sentence to "The green area denote the river basin".

Response to comment No. 9: Thank you very much for your comment. According to the comment, "green coloured shape denotes the basin boundary" has been changed to "green area denotes the river basin". Line 802

10. Please modify the y-label of Figure 8. It looks like number of days divided by 25 years.

Response to comment No. 10: Thank you very much for your comment. According to the comment, the y-label of Figure 8 has been modified to avoid confusion. Line 823

11. Supplement Line 150: Please add the unit of cropping intensity.

Response to comment No. 11: Thank you very much for your comment. As the reviewer knows, the cropping intensity can be explained as the number of times the crops are grown on a given agricultural land within a year. Cropping intensity of 1.5 means that on average 150% of the total irrigated cropland is used for cultivation (i.e., 100% of the land is used for cultivation of the first crop and 50% of the land is used for cultivation of the second crop). Since the cropping intensity is unitless, these explanations were added in the text for more clarity as follows:

*"Cropping intensity of 1.5 means that on average 150% of the total irrigated cropland is used for cultivation."* Line 170

---

## Author Response (AR2)

**Hydrology and Earth System Sciences**

**Inclusion of flood diversion canal operation in the H08 hydrological model with a case study from the Chao Phraya River Basin – Part 1: Model development and validation**

(hess-2021-532)

Saritha Padiyedath Gopalan, Adisorn Champathong, Thada Sukhapunnaphan, Shinichiro Nakamura, and Naota Hanasaki

......................................................

**COMMENTS FROM EDITOR**

......................................................

Dear Authors,

both Referees are very satisfied with your revised manuscript.

Ref#1 has submitted a few additional comments, suggesting some technical corrections based on a few remaining doubts on the procedure. I invite you to try to address also such points, since the clarifications asked by referee will certainly be beneficial also to other readers who may have the same doubts.

Many thanks again for having submitted your interesting work to HESS,

best wishes,

Elena Toth

**Response to Editors comments**

Dear Editor,

Thank you very much for the positive editorial decision. We are also thankful to both the reviewers for assessing and evaluating the revised version of our manuscript. Their constructive reviews for the improvement of the manuscript are highly appreciated.

We have addressed the three additional questions raised by Reviewer 1 and the manuscript has carefully revised according to these questions. The text that has been added or modified was underlined in the revised manuscript. Also, we have utilized the note tool of Microsoft Word

to add comments to the corresponding changes made by the authors. The following are the reviewer's comments and the corresponding point-by-point replies.

**Comments from Reviewer #1:**

Many thanks to the authors for their serious response to reviewers' comments. The revised manuscript has greatly improved in clarity. The added information in Table S1 is especially helpful for understanding the results in Figure 5. I now only have three additional questions where clarification from the authors would be appreciated, and I look forward to seeing the paper published soon.

**Reply:** Thank you very much for your positive feedback on our revised manuscript. We highly appreciate your review comments that provided valuable insights for further modifications of the initial version of our manuscript.

The detailed point-by-point replies to all your concerns are given below. Once again thank you for enlightening us with your valuable comments and suggestions.

**Comments**

1. Based on Table S1, REG/GEN parameters are not that different for Chainat-Pasak, are the large differences in the second half of the year related to the excessive canal discharge under GEN simulation at Makham Thao - Uthong?

Thank you very much for your comment. We agree with your point that the large differences in the diverted canal flow during the second half of the year is due to the increased canal carrying capacity under the GEN scheme for the Makham Thao – Uthong canal. The canal carrying capacity values under the GEN scheme are very high compared with the OBS and REG scheme values (Table S1). This is because the primary purpose of all canals under the GEN scheme is flood control (because flood control is the primary objective of this study) and fixes the canal carrying capacity at the $Q_{50}$ value, leading to the overestimation of canal flow under this scheme. This GEN scheme can be improved by differentiating the purpose of each canal in the simulations, similar to the approach in the REG simulation. We have already incorporated such discussion in the manuscript (Section 4.4; Line 520-530).

2. Is there any uncertainty associated with the observed carrying capacities, which are solely determined by their cross-sections (Line 285)? This discharge is determined by cross

section area multiplied by flow speed right? While the former can be measured with good accuracy, how is the maximum flow speed before flooding estimated, and does this change with time?

Thank you very much for your comment. Before addressing the comment, we would like to clarify that the observed carrying capacities were not computed by the authors, instead these are the values collected by literature review (JICA, 2013; Tamada et al., 2013) as well as from the Royal Irrigation Department, Thailand. Of course, there will be uncertainty associated with the observed carrying capacities because the maximum flow rate varies over time. However, for this modelling purpose, we have used the available observed data based on the review of literature. To avoid confusion, we have clearly mentioned in the manuscript that the observed values were accessed from the existing literature. Line 285

JICA (Japan International Cooperation Agency): Project for the comprehensive flood management plan for the Chao Phraya River basin, Final report, Volume 1: Summary Report, CTI Engineering International Co., Ltd., Oriental Consultants Co., Ltd., Nippon Koei Co., Ltd., CTI Engineering Co., Ltd., available at: https://openjicareport.jica.go.jp/pdf/12127205.pdf, September 2013.

Tamada, Y., Hoshikawa, K. and Funatsu, T. (Eds.): The 2011 Thailand Floods: Lessons and Records, Josei-Bunseki Report No.22, Institute of Developing Economies, IDE-JETRO, Japan, 2013.

3. I now understand better "immediate neighboring cell" thanks to updated Figure 2 and corresponding text. So a total of 20% discharge is supplied (10% to the cell that canal flows through, and 10% to the other nearest cell) each time canal flow through a grid right? Then if the initial discharge is 100, after one cell it becomes 80, and after next cell it becomes 64? I am not quite sure how to identify the immediate neighboring cell when the flow first turns toward southeast (Figure 2): it seems the cell on its bottom is equally close to the cell on its right, do you compute the Euclidean distance of each neighboring cell to canal vector in order to determine the immediate neighbor?

Thank you very much for your comment. The authors are happy to know that our explanations enhanced your understanding regarding the "immediate neighbouring cell". We would like to clarify that the immediate neighbouring cells are decided based on the presence of croplands, not by the distance from the canal. If the immediate neighbouring grid cells are croplands (both

left and right of the canal), then 10% of the canal water is supplied to each of these grid cells and a total of 30% canal discharge is supplied each time the canal water flow through a grid. Conversely, if no cropland was identified in the immediate grid cell, only 10% of discharge is supplied to the cell that canal flows through. We are extremely sorry for not clarifying this point in the revised manuscript. Based on this comment, we have added these explanations to the revised manuscript and once again thank you very much for reminding this point. Line 192